# A Bayesian Approach for Personalized Federated Learning in Heterogeneous Settings

## Abstract

In several practical applications of federated learning (FL), the clients are highly heterogeneous in terms of both their data and compute resources, and therefore enforcing the same model architecture for each client is very limiting. The need for uncertainty quantification is also often particularly amplified for clients that have limited local data. This paper presents a unified FL framework based on training customized local Bayesian models that can simultaneously address both these constraints. A Bayesian framework provides a natural way of incorporating supervision in the form of prior distributions. We use priors in the functional (output) space of the networks to facilitate collaboration across heterogeneous clients via an unlabelled auxiliary dataset. We further present a differentially private version of the algorithm along with formal differential privacy guarantees that apply to general settings without any assumptions on the learning algorithm. Experiments on standard FL datasets demonstrate that our approach outperforms strong baselines in both homogeneous and heterogeneous settings and under strict privacy constraints, while also providing characterizations of model uncertainties.

## 1 Introduction

The vast majority of research on Federated Learning (FL) takes an optimization perspective i.e., focus on designing suitable optimization objectives or provide methods for efficiently solving such objectives. However, there are several critical applications where other issues such as obtaining good estimates of the uncertainty in an outcome, or data privacy constraints, are equally pressing. For example, in certain healthcare applications, patient data is private and uncertainty around prediction of health outcomes is required to manage risks. Similarly, for several applications in legal, finance, mission critical IoT systems (Victor et al., 2022), etc., both the privacy and confidence guarantees are important for decision making.

Since Bayesian learning methods are known for their ability to generalize under limited data, and can generally provide well-calibrated outputs (Wilson et al., 2016; Achituve et al., 2021a;b), it makes them suitable for solving the challenges mentioned above. A straightforward Bayesian FL method would perform the following steps - each client does local posterior inference to obtain a distribution over weight parameters and then communicates the local posteriors to the server, the server receives the local posteriors from the clients and aggregates them to obtain a global posterior distribution which is then broadcast to the clients for the next round of training. However, this entire learning procedure is highly resource and communication-intensive. For a very simple example, solving an $m$-dimensional federated least squares estimation, this method will require $O(m^3)$ computation on all the clients and server sites (Al-Shedivat et al., 2021). This cost is much more as opposed to the standard FL which is generally $O(m)$. How could the Bayesian methods be then used for FL settings without paying such high costs? Also, since clients can have varied degrees of computing resources, constraining each client to train identical models would be limiting. Then, how to aggregate the local posteriors when the clients are training personal models and the model weights across clients are not identical, becomes the next important question. We try to address these questions by proposing a framework that allows all clients to train their own personal Bayesian models (with varying model complexities), but still enables collaboration across clients by instilling information from the peer collaboration in the form of priors. This is achieved by using the functional space to indirectly determine priors on weights, as opposed to them being specified in the weight space as in traditional Bayesian methods.

In addition, maintaining client data privacy is an important concern in FL. While attacks on neural networks that can recover training data from the models by either directly analyzing model parameters or indirectly by analyzing the model outputs are well known (Fredrikson et al., 2015; Papernot et al., 2017), even in federated learning settings, when the data is restricted to the local client sites, it has been shown that FL methods are not immune to data leaks (Fowl et al., 2021; Wen et al., 2022; Fowl et al., 2023). Therefore, to guarantee the privacy of the local client data, we apply a formal well-known standard of differential privacy (Dwork & Roth, 2014). We use a carefully designed noise mechanism for data sharing that allows us to provide a privacy bound on the entire procedure.

In this work, we propose a novel unified FL framework that can simultaneously address challenges arising due to limited data per client and heterogeneous compute resources across clients, together with the need for privacy guarantees and providing calibrated predictions. To the best of our knowledge, no previous work has jointly addressed all these learning issues in the FL context. Our positive results substantially increase the scope of FL for critical real-world applications.

**Our key contributions** are summarized as follows :

1. We propose a novel computation and communication efficient method for personalized federated learning based on Bayesian inference. In this method, collaboration across clients is achieved through a novel way of assigning prior distributions over the model parameters via the output space. We call this method Federated Bayesian Neural Networks (FedBNN).

2. We present a non-trivial extension of our method that can work with heterogeneous settings where clients are not forced to train models of identical architectures. This provides more flexibility to the clients when the compute resources across clients are not identical or clients have existing pre-trained solutions.

3. We provide a formal differential privacy guarantee for our method that applies to general settings irrespective of the client's learning algorithm and show that the method is able to learn effectively even under strict privacy guarantees.

4. We evaluate our method on several datasets and show that it outperforms the baselines by a significant margin, particularly in data and model heterogeneous settings, making it particularly more useful for FL applications where a high degree of heterogeneity is naturally present.

## 2 Related Work

This section provides an overview of the most relevant prior work in the fields of federated learning, Bayesian FL, and Differential Privacy in FL.

**Federated Learning** In the early 2000s, privacy preserving distributed data mining referred to training distributed machine learning models (Kargupta & Park, 2000; Gan et al., 2017; Aggarwal & Yu, 2008), like distributed clustering (Merugu & Ghosh, Nov, 2003; 2005), distributed PCA (Kargupta et al., 2001), distributed SVMs (Yu et al., 2006) etc. Federated Learning was introduced as the FedAvg algorithm in the seminal work by (McMahan et al., 2017b). Since then many different modifications have been proposed that tackle specific challenges. FedPD (Zhang et al., 2021), FedSplit (Pathak & Wainwright, 2020), and FedDyn (Acar et al., 2021) proposed methods for finding better fixed-point solutions to the FL optimization problem. (Lin et al., 2020; Yurochkin et al., 2019; Wang et al., 2020; Singh & Jaggi, 2020; Chen & Chao, 2021) show that point-wise aggregate of the local client models does not produce a good global model and propose alternate aggregation mechanisms to achieve collaboration. FedDF (Lin et al., 2020) and (Li & Wang, 2019; Ozkara et al., 2021) achieve personalised FL solutions through collaboration by performing knowledge distillation on local client models. Personalised FL has been approached in many other ways like meta-learning (Fallah et al., 2020; Beaussart et al., 2021; Jiang et al., 2019; Khodak et al., 2019), multi-task learning (Smith et al., 2017a; Li et al., 2021; Smith et al., 2017b), by clustering the clients (Sattler et al., 2021b; Ghosh et al., 2020) and others (Collins et al., 2021; Li et al., 2020a; Yu et al., 2020; Shamsian et al., 2021), (Wang et al., 2023; Makhija et al., 2022), (Chen et al., 2022) uses Bayesian view based analysis to obtain a better trade-off between personal and global models. The personalised FL methods focus on improving the performance in the presence of statistical data heterogeneity across clients, but they do not work well when the size of dataset on the clients is limited.

**Bayesian Federated Learning** The Bayesian approaches for federated learning can be broadly divided into two categories - methods using Bayesian inference for local learning at each client and methods that achieve collaboration via Bayesian mechanisms. Amongst the methods that use Bayesian approach for achieving collaboration across clients, FedBE (Chen & Chao, 2021) uses Bayesian mechanism to aggregate the locally trained neural networks to obtain a Bayesian ensemble at the server. (Bhatt et al., 2022) suggests using knowledge distillation and MCMC based method for training a global model. (Dai et al., 2020) on the other hand suggests the use of Bayesian Optimization and Thompson Sampling to obtain the solution to the global optimization problem. PFNM (Yurochkin et al., 2019) uses a Beta-Bernoullli process to obtain a global model from the local models but is only applicable to fully-connected networks, FedMA (Wang et al., 2020) extends PFNM for other types of networks. Recently, (Ozer et al., 2022) did an empirical study on various ways of aggregating variational Bayesian neural networks and their effects. On the other hand, FedPA (Al-Shedivat et al., 2021) was the first to use local Bayesian inference and suggested that the global posterior distribution of weights could be obtained by multiplying local posteriors and proposed an approximate and efficient way of computing local and global posteriors by using Laplace approximations with complexity that is linear in number of model parameters. This method focuses on obtaining a global solution and is less suited for the statistical heterogeneity present across clients (Cao et al., 2023), and therefore we focus more on the methods that build personalized solutions for clients. Among such methods, pFedGP (Achituve et al., 2021b) is a Gaussian Process-based estimation method where the kernel in these GPs is defined on the output of a neural network and is called Deep Kernel Learning. pFedGP works by collaboratively training a single deep kernel for all clients but using personalized GPs for prediction. The collaboration across clients while learning the global kernel is done as in FedAvg. FedLoc (Yin et al., 2020) also uses GP in FL but for regression tasks. pFedBayes (Zhang et al., 2022) uses variational inference for local posterior inference on Bayesian Neural Networks(BNNs) where the loss at each client is a combination of the data likelihood term and distance to the prior where prior distribution is replaced by the global distribution. The global distribution is obtained by aggregating the local prior distributions. FOLA (Liu et al., 2021) proposed using Laplace Approximation for posterior inference at both the server side and the client side. (Ozkara et al., 2023) and (Kotelevskii et al., 2022) also proposed methods that assume a hierarchical Bayesian model on the parameters but their main assumption is that the local parameters are obtained from the same distribution thus making them useful only in homogeneous settings. None of the methods described above explicitly handle heterogeneous settings. Moreover, for these methods choosing an appropriate prior distribution over the local model parameters is a challenge (Cao et al., 2023). These issues led us to use functional space priors instead. Such priors have been studied in limited centralized settings (Tran et al., 2022; Sun et al., 2019; Flam-Shepherd, 2017) but not in FL settings.

**Differential Privacy in FL** Since decentralized learning does not guarantee that the data will remain private, it is important that a formal rigorous guarantee be given on the data that is leaked by the algorithm. Seminal works in DP propose using a Gaussian noise mechanism by adding Gaussian noise to the intermediate results and achieving a bound on the algorithm by using composition results (Dwork & Roth, 2014; Mironov, 2017; Kairouz et al., 2015). For FL, (Geyer et al., 2017) and (McMahan et al., 2017a) independently proposed DP-FedSGD and DP-FedAvg algorithms which enhance FedAvg by adding Gaussian noise to the local client updates. Several other works focus on analyzing the privacy-utility trade-off in DP in FL setting (Ghazi et al., 2019; Girgis et al., 2021; Balle et al., 2019; Triastcyn & Faltings, 2019; Li et al., 2020b). Recently, (Hu et al., 2020) proposed a DP-based solution for personalized FL which only works for linear models. And then (Noble et al., 2022) enhanced it for general models and heterogeneous data in FL. These methods, however, mostly focus on privacy guarantees while solving the FL optimization problem. They don't apply to general Bayesian settings or when substantial heterogeneity is encountered.

## 3 PROPOSED METHODOLOGY

We propose a novel method for Bayesian FL that enables privacy preserving personalised learning on clients under heterogeneous settings. The local training is done in a model agnostic way but collaboration amongst clients is enabled by passing information as the prior. And to achieve privacy, like most private algorithms, we use Gaussian mechanism to carefully add noise to the information

sent outside a client. This method also produces well calibrated outputs. In this section, we first describe the problem statement and then go over the proposed solution.

## 3.1 PROBLEM DESCRIPTION

Consider an FL setting with $N$ clients where each client $i$ has local dataset $\mathcal{X}_i$ of size $n_i$ drawn from the local data distribution $\boldsymbol{D}_i$. The goal of a personalised federated learning procedure is to obtain optimal weights for each client's local model, $\mathcal{W}_i$, using data $\mathcal{X} = \bigcup_{j=1}^{N} \mathcal{X}_j$ without actually accessing the data outside of each client, i.e., no other client can access any data in $\mathcal{X}_i$ except the $i^{th}$ client itself but the clients could transfer knowledge via collaboration. In a personalised Bayesian learning procedure, the modified goal would be to learn distribution over local weights, $\mathbb{P}(\mathcal{W}_i|\mathcal{X})$ from $\mathcal{X} = \bigcup_{j=1}^{N} \mathcal{X}_j$ while still maintaining the client data privacy. However, the learning procedure faces challenges that are posed due to - *system heterogeneity* and *statistical heterogeneity*. System heterogeneity refers to the variable amount of data and compute resources across clients, meaning, i) the data resources on each client vary widely, i.e., $n_k >> n_l$ for some clients $k$ and $l$, and ii) the compute across clients is non-identical due to which it is not possible to train models of uniform architectures across clients, leading to non-identical weights, i.e., $\mathcal{W}_i \neq \mathcal{W}_j$ for different clients $i$ and $j$. Statistical heterogeneity implies that the data distribution across clients is non-IID.

## 3.2 FEDBNN METHODOLOGY

Here we propose an adaptive framework to learn personalised Bayesian Neural Network (BNN) based models for each client that can work effectively even under the challenges described above. The overall framework works iteratively in two steps - local optimization on the individual client to obtain local posterior distribution over the model parameters, and a global collaboration step where the output from each client is appropriately aggregated at the server and broadcast to all the clients for the next rounds of training. Each of these two steps is described in detail below, and the detailed algorithm is given in the Appendix B.

**Local Setting** Let each client in the network be training a personalised Bayesian NN, which for the client $i$ is denoted by $\Phi_i$ and is parameterised by weights $\mathcal{W}_i$. As commonly used in the literature, we assume that the individual weights of the BNN are Normally distributed and satisfy mean-field decomposition, i.e., $w_{i,\alpha} \sim \mathcal{N}(\mu_{i,\alpha}, \sigma_{i,\alpha}^2)$ for $\alpha \in [1, \ldots, |\mathcal{W}_i|]$ where $\mu_{i,\alpha}$ is the mean of the gaussian distribution for the parameter $\alpha$ on the $i^{th}$ client and $\sigma_{i,\alpha}^2$ is the variance of the gaussian for the same parameter. To guarantee that $\sigma_{i,\alpha}$ takes non-negative values for all clients $i$ and all parameters $\alpha$, we use a technique commonly used in inference procedures (Blundell et al., 2015), wherein each $\sigma_{i,\alpha}$ is replaced by another parameter $\rho_{i,\alpha}$ during the training, such that $\sigma_{i,\alpha} = \log(1 + \exp(\rho_{i,\alpha}))$.

### 3.2.1 GLOBAL COLLABORATION

We attain collaboration amongst clients under a mild assumption of availability of a general publicly accessible unlabelled dataset at the server. We call this dataset as Alignment Dataset (AD). This dataset is used for providing peer supervision to the clients by helping clients distill knowledge from other clients without sharing the model weight distributions. While in heterogeneous settings non-identical architecture models across clients mean no direct way of aggregation and prior specification, even in homogeneous settings aggregating the weight distributions and specifying priors in the weight space is prone to errors due to reasons like non-alignment of weights across clients, insufficient understanding of the weight space, etc. Thus, for the purpose of collaboration, we move away from the weight-space to the function-space of the networks. Specifically, in each global communication round, the server shares the AD with all the clients. The clients do a forward pass on AD to obtain the output $\Phi_i(AD)$. The local output of the $i^{th}$ client is approximated by doing Monte Carlo sampling and drawing $K$ weight samples, $\mathcal{W}_i^{(j)} : j \in [1, K]$, from its local posterior distribution $\mathbb{P}(\mathcal{W}_i|\mathcal{X})$. An aggregate of the obtained logits for these $K$ weight samples under the client's own personal BNN model, $\Phi_i()$, is reported, i.e. $\Phi_i(AD) = \frac{1}{K} \sum_{j=1}^{K} \Phi_i(AD; \mathcal{W}_i^{(j)})$. The obtained output for AD on each client is then sent back to server which forms an aggregated representation, denoted by $\bar{\Phi}(AD)$, by doing a weighted aggregation of all clients' outputs, i.e., $\bar{\Phi}(\mathbf{X}) = \sum_{j=1}^{N} w_j \Phi_j(\mathbf{X})$. The weights

$w$'s used in the aggregation could represent the strength of that particular client in terms of its data or compute resources, i.e., clients with high compute (or data) resources receive more weight as compared to clients with lower amount of resources. The obtained $\bar{\Phi}(\text{AD})$ is then uploaded to all the clients for use in the next round of local training. More details about the Alignment Dataset (AD) are included in the Appendix E.

### 3.2.2 LOCAL OPTIMIZATION ON CLIENTS

**Prior Specification**  The Bayesian framework provides a natural way of incorporating supervision in the form of priors. Conventional methods in Bayesian deep learning provide priors over model parameters. However, the relationship between the model parameters and the outputs is complex and the priors in model's weight-space do not directly capture the desired functional properties. Also, since the number of parameters in a neural network is large, most prior specifications tend to take a simplistic form like an isotropic Gaussian, to make inference feasible. Thus, learning by specifying prior distributions over weights does not always help incorporate prior knowledge in the learning process. In this work, we consider a way of specifying priors in the functional space by first optimising the Bayesian neural networks over the prior parameters for a fixed number of steps to achieve a desired functional output. While being more intuitive, these priors also help in instilling the prior external knowledge during the training of the neural networks.

**Local Optimization**  For the local optimization, the individual clients learn $\mathbb{P}(\mathcal{W}_i|\mathcal{X}_i)$ via variational inference. A variational learning algorithm, tries to find optimal parameters $\theta^*$ of a parameterized distribution $q(\mathcal{W}_i|\theta)$ among a family of distributions denoted by $\mathcal{Q}$ such that the distance between $q(\mathcal{W}_i|\theta^*)$ and the true posterior $\mathbb{P}(\mathcal{W}_i|\mathcal{X}_i)$ is minimized. In our setting, we set the family of distributions, $\mathcal{Q}$, to be containing distributions of the form $w_{i,\alpha} \sim \mathcal{N}(\mu_{i,\alpha}, \sigma_{i,\alpha}^2)$ for each parameter $w_{i,\alpha}$ for $\alpha \in [1, \ldots, |\mathcal{W}_i|]$. Let $p(\mathcal{W}_i; \psi)$ represent the prior function over the weights $\mathcal{W}_i$ and is parameterized by $\psi$, then the optimization problem for local variational inference is given by :

$$\theta_i^* = \underset{\theta : q(\mathcal{W}_i|\theta) \in \mathcal{Q}}{\arg\min} \ \text{KL}[q(\mathcal{W}_i|\theta)||\mathbb{P}(\mathcal{W}_i|\mathcal{X}_i)] \tag{1}$$

$$= \underset{\theta : q(\mathcal{W}_i|\theta) \in \mathcal{Q}}{\arg\min} \ \text{KL}[q(\mathcal{W}_i|\theta)||p(\mathcal{W}_i; \psi)] - \mathbb{E}_{q(\mathcal{W}_i|\theta)}[\log \mathbb{P}(\mathcal{X}_i|\mathcal{W}_i)]. \tag{2}$$

For inference in Bayesian neural networks, we use Bayes by Backprop (Blundell et al., 2015) method to solve the variational inference optimization problem formulated above.

At the beginning of each local optimization procedure (in each global communication round a specific client is selected), we use the global information obtained from the server $\bar{\Phi}(\text{AD})$ to intialize the prior for the BNN. Specifically, at the beginning of each local training round, the selected clients first tune their priors to minimize the distance between the local output, $\Phi_i(\textbf{AD}; \mathcal{W}_i)$ and the aggregated output obtained from the server, $\bar{\Phi}(\text{AD})$. Since the aggregated output represents the collective knowledge of all the clients and may not be *strictly precise* for the local model optimization, we consider this aggregated output as "noisy" and correct it before using for optimization. Specifically, we generate $\Phi_i^{\text{corrected}}$ as a convex combination of the global output and the local output for a tunable parameter $\gamma$. For the $i^{th}$ client,

$$\Phi_i^{\text{corrected}} = \gamma \bar{\Phi}(\text{AD}) + (1 - \gamma)\Phi_i(\text{AD}; \mathcal{W}_i). \tag{3}$$

The prior optimization steps then optimize the distance between $\Phi_i^{\text{corrected}}$ and $\Phi_i(\text{AD}; \mathcal{W}_i)$ to train the prior parameters $\psi$, with the aim of transferring the global knowledge encoded in $\Phi_i^{\text{corrected}}$ to the local model. Precisely,

$$\psi_i^* = \underset{\psi}{\arg\min} \, \text{d}(\Phi_i^{\text{corrected}}, \Phi_i(\text{AD}; \mathcal{W}_i)). \tag{4}$$

When the outputs $\Phi(\text{X}; \mathcal{W})$ are logits, we use cross-entropy or the negative log-likelihood loss as the distance measure. The optimization involves training the client's personal BNN $\Phi_i$ to only learn the parameters of the prior distribution denoted by $\psi$. This way of initializing the BNN prior enables translating the functional properties, as captured by $\Phi_i(\text{AD}; \mathcal{W}_i)$, to weight-space distributions. The optimal prior parameters are then kept fixed while training the BNN over the local dataset. The local optimization procedure now works to find the best $q(\mathcal{W}_i|\theta)$ fixing the prior distribution through the

following optimization problem :

$$\theta_i^* = \underset{\theta:q(\mathcal{W}_i|\theta)\in\mathcal{Q}}{\arg\min} \ \text{KL}[q(\mathcal{W}_i|\theta)||p(\mathcal{W}_i;\psi_i^*)] - \mathbb{E}_{q(\mathcal{W}_i|\theta)}[log\mathbb{P}(\mathcal{X}_i|\mathcal{W}_i)]. \tag{5}$$

### 3.2.3 Achieving Differential Privacy

In this method, we measure the privacy loss at each client. To control the release of information from the clients, we add a carefully designed Gaussian mechanism wherein we add Gaussian noise to the $\Phi_i(\text{AD})$ that is being shared by each client. Specifically, each client $i$ uploads $\Phi_i(\text{AD})_{\text{DP}} = \Phi_i(\text{AD}) + \mathcal{N}(0, \sigma_g^2)$ to the server and then the server aggregates $\Phi_i(\text{AD})_{\text{DP}}$ across clients to obtain and broadcast $\bar{\Phi}(\text{AD})_{\text{DP}}$ which is used by the clients in their next round of local optimization. The variance of the noise depends on the required privacy guarantee.

## 4 Privacy Analysis

Since our algorithm does not specifically share model weights outside of the clients but shares the client models' output on a public dataset, it might seem intuitive that the algorithm is private but knowing that privacy is lost in unexpected ways, we present a formal Differential Privacy based guarantee on the algorithm. Our analysis in this section focuses on providing record-level DP guarantee over the entire dataset $\mathcal{X}$. This analysis quantifies the level of privacy achieved towards any third party and an honest-but-curious server. In this section we first define DP and then present our analysis. Additional definitions and the results used in the proof are mentioned in the Appendix C.

**Definition 4.1** (($\epsilon, \delta$)- Differential Privacy). A randomized algorithm $\mathcal{M}$ is ($\epsilon, \delta$)-DP if for any two neighboring datasets $D$ and $D'$ that differ in at most one data point, the output of the algorithm $\mathcal{M}$ on $D$ and $D'$ is bounded as

$$\mathbb{P}[\mathcal{M}(D) \in S] \leq e^\epsilon \mathbb{P}[\mathcal{M}(D') \in S] + \delta, \quad \forall S \subseteq \text{Range}(\mathcal{M}).$$

Due to the lack of space, definitions of zero-concentrated differential privacy (zCDP), $L_2$ Sensitivity of an algorithm, and results corresponding to Gaussian Mechanism, Sequential Composition and Parallel Composition of privacy are given in Appendix C. Building on top of these results, we now state the privacy budget of our algorithm.

**Theorem 4.2** (Privacy Budget). *The proposed algorithm is ($\epsilon, \delta$)-differentially private, if the total privacy budget per global communication round per query is set to*

$$\rho = \frac{\epsilon^2}{4EK log\frac{1}{\delta}}$$

*for $E$ number of global communication rounds and $K$ number of queries to the algorithm per round.*

*Proof.* After using Gaussian mechanism on each client and adding noise to each coordinate of $\Phi_i(\text{AD})$, the local mechanism at each client becomes $\rho$-zCDP for $\rho = \frac{\Delta^2}{2\sigma^2}$. Since each client outputs the logit representation for each input, i.e., the normalized output of the clients, $\Delta^2 \leq 2$. Suppose in each global communication round we make $K$ queries to each client, then by sequential composition C.2, we get $EK\rho$, for $E$ number of global communication rounds. By parallel composition C.3, the total privacy loss for all $N$ clients is the maximum of the loss on each client and therefore remains $EK\rho$. Relating it to the ($\epsilon, \delta$)-DP from C.1, we get $\rho = \dfrac{\epsilon^2}{4EK\log\frac{1}{\delta}}$ for any $\delta > 0$. $\qquad\square$

Our analysis does not assume any specifics of how each client is trained and is therefore applicable in more general settings. However, we present a pessimistic analysis by providing a worst-case analytical bound. This is because we assume that a change in single data point may entirely change the output of the algorithm and upper bound the $L_2$ sensitivity $\Delta^2 \leq 2$, and also, since the public dataset remains common throughout the rounds, the actual privacy loss due to querying on the public dataset will not typically add up linearly. Yet the above analysis shows that we have several knobs to control to achieve the desired privacy-utility trade off - balancing the number of global communication rounds with local epochs, reducing the number of queries, and the standard noise scale. By appropriately tuning these controls we are able to achieve good performance with a *single digit $\epsilon$ ($\approx 9.98$)* privacy guarantee and $\delta = 10^{-4}$.

# 5 EXPERIMENTS

In this section, we present an experimental evaluation of our method and compare it with different baselines under diverse homogeneous and heterogeneous client settings. Specifically, we experiment with three types of heterogeneity - i) heterogeneity in data resources (amount of data), ii) heterogeneity in compute resources, and iii) statistical heterogeneity (non-IID data distribution across clients). We also discuss the change in performance of our method when the degree and type of heterogeneity changes. Due to the space constraint, additional experiments on varying the size and distribution of the AD, privacy-utility trade-off and showing calibrated outputs are included in the Appendix E, G and D respectively.

## 5.1 EXPERIMENTAL DETAILS

**Datasets** We choose three different datasets commonly used in prior federated learning works from the popular FL benchmark, LEAF (Caldas et al., 2019) including MNIST, CIFAR-10 and CIFAR-100. MNIST contains 10 different classes corresponding to the 10 digits with 50,000 $28 \times 28$ black and white train images and 10,000 images for validation. CIFAR-10 and CIFAR-100 contain 50,000 train and 10,000 test-colored images for 10 classes and 100 classes respectively. The choice of these datasets is primarily motivated by their use in the baseline methods.

**Simulation Details** We simulate three different types of heterogeneous settings - corresponding to heterogeneity in compute resources, data resources and the statistical data distribution. Before starting the training process, we create $N$ different clients with different compute resources by randomly selecting a fraction of clients that represent clients with smaller compute. Since these clients do not have large memory and compute capacity, we assume that these clients train smaller-size BNNs as opposed to the other high-capacity clients that train larger VGG-based models. In particular, the small BNNs were constructed to have either 2 or 3 convolution layers, each followed by a ReLU and 2 fully-connected layers at the end, and a VGG9-based architecture was used for larger BNNs. The number of parameters in smaller networks is around 50K and that in larger networks is around 3M. Since the baselines only operate with identical model architectures across clients, we use the larger VGG9-based models on the baselines for a fair comparison. We include the results of our method in both homogeneous compute settings (similar to baselines) as well as in heterogeneous compute settings wherein we assume that 30% of the total clients have smaller compute and are training smaller-sized models.

Next, we also vary the data resources across clients and test the methods under 3 different data settings - small, medium and full. The small setting corresponds to each client having only 50 training data instances per class, for the medium and full settings each client has 100 data instances and 2500 data instances per class respectively for training. We simulate statistical heterogeneity by creating non-IID data partitions across clients. We work in a rather strict non-IID setting by assuming clients have access to data of disjoint classes. For each client a fraction of instance classes is sampled and then instances corresponding to the selected classes are divided amongst the specific clients. For the included experiments, we set number of clients $N = 20$ and divide the instances on clients such that each client has access to only 5 of the 10 classes for MNIST and CIFAR-10, and 20 out of 100 classes for CIFAR-100.

**Training parameters and Evaluation** We run all the algorithms for 200 global communication rounds and report the accuracy on the test dataset at the end of the $200^{th}$ round. The number of local epochs is set to 20 and the size of AD is kept as 2000. Each client is allowed to train its personal model for a fixed number of epochs, which is kept to 50 in experiments, before entering the collaboration phase. The hyper-parameters of the training procedure are tuned on a set-aside validation set. At the beginning of each global communication round, for optimizing the prior parameters at each client according to Equation 4, we use an Adam optimizer with learning rate=0.0001 and run the prior optimization procedure for 100 steps. Then with the optimized prior we train the local BNN using Bayes-by-Backprop, with Adam optimizer, learning rate = 0.001 and batch size = 128. The noise effect $\gamma$ is selected after fine-tuning and kept to be 0.7. For these experiments, the aggregation weight $w_j$ for each client $j$ used to compute $\bar{\Phi}(\mathbf{X})$ is set to $1/N$. All the models are trained on a 4 GPU machine with GeForce RTX 3090 GPUs and 24GB per GPU memory. For evaluation, we report

Table 1: Test accuracy comparsion with baselines in non-IID settings.

| Method | MNIST | | | CIFAR10 | | | CIFAR100 | | |
|---|---|---|---|---|---|---|---|---|---|
| | (small) | (medium) | (full) | (small) | (medium) | (full) | (small) | (medium) | (full) |
| **(Non-Bayesian)** | | | | | | | | | |
| Local Training | $88.7 \pm 1.2$ | $90.1 \pm 1.0$ | $91.9 \pm 1.1$ | $53.9 \pm 2.1$ | $59.5 \pm 1.8$ | $70.8 \pm 1.4$ | $28.8 \pm 1.8$ | $32.7 \pm 1.9$ | $43.5 \pm 1.6$ |
| FedAvg | $88.2 \pm 0.5$ | $90.15 \pm 1.2$ | $92.23 \pm 1.1$ | $43.14 \pm 1.2$ | $56.27 \pm 1.8$ | $78.17 \pm 1.2$ | $27.3 \pm 1.9$ | $32.81 \pm 1.6$ | $36.3 \pm 0.2$ |
| FedProx | $86.9 \pm 0.8$ | $89.91 \pm 0.7$ | $93.1 \pm 0.4$ | $44.27 \pm 1.2$ | $58.93 \pm 0.9$ | $79.19 \pm 0.6$ | $28.6 \pm 2.7$ | $34.31 \pm 1.4$ | $37.8 \pm 0.9$ |
| pFedME | $91.95 \pm 2.1$ | $93.39 \pm 1.2$ | $95.62 \pm 0.5$ | $48.46 \pm 1.5$ | $64.57 \pm 2.1$ | $75.11 \pm 1.2$ | $32.4 \pm 2.2$ | $36.3 \pm 2.0$ | $41.8 \pm 1.7$ |
| KD based collaboration | $89.1 \pm 0.4$ | $92.5 \pm 0.2$ | $93.2 \pm 0.3$ | $33.9 \pm 1.3$ | $53.2 \pm 1.5$ | $69.8 \pm 1.0$ | $26.1 \pm 2.0$ | $35.2 \pm 1.2$ | $42.7 \pm 0.8$ |
| **(Bayesian with Homogeneous Architectures)** | | | | | | | | | |
| pFedGP | $86.15 \pm 1.3$ | $90.59 \pm 1.7$ | $94.92 \pm 0.3$ | $45.62 \pm 2.2$ | $56.24 \pm 1.8$ | $72.89 \pm 0.7$ | $47.06 \pm 1.3$ | $53.1 \pm 1.2$ | $54.54 \pm 0.2$ |
| pFedBayes | $94.0 \pm 0.2$ | $94.6 \pm 0.1$ | $95.5 \pm 0.3$ | $58.7 \pm 1.1$ | $64.6 \pm 0.8$ | $78.3 \pm 0.5$ | $39.51 \pm 1.8$ | $41.43 \pm 0.4$ | $47.67 \pm 1.1$ |
| FOLA | $91.74 \pm 1.0$ | $92.87 \pm 0.8$ | $95.12 \pm 0.6$ | $43.29 \pm 0.9$ | $45.94 \pm 0.7$ | $67.98 \pm 0.5$ | $33.42 \pm 1.3$ | $48.8 \pm 2.1$ | $43.2 \pm 1.6$ |
| Ours (Homo) | $\mathbf{94.9 \pm 1.0}$ | $\mathbf{95.72 \pm 0.8}$ | $\mathbf{96.21 \pm 0.3}$ | $\mathbf{70.6 \pm 1.1}$ | $\mathbf{72.3 \pm 0.6}$ | $\mathbf{79.7 \pm 0.3}$ | $\mathbf{49.65 \pm 1.4}$ | $\mathbf{55.4 \pm 0.8}$ | $\mathbf{57.3 \pm 0.8}$ |
| Ours (Hetero) | $93.1 \pm 1.1$ | $94.4 \pm 0.2$ | $95.9 \pm 0.2$ | $\mathbf{68.17 \pm 2.0}$ | $\mathbf{71.73 \pm 1.3}$ | $\mathbf{78.7 \pm 0.7}$ | $47.5 \pm 1.4$ | $49.10 \pm 1.1$ | $\mathbf{51.1 \pm 0.7}$ |
| Ours (Hetero-DP) | $89.82 \pm 2.3$ | $90.21 \pm 1.6$ | $91.43 \pm 1.4$ | $60.4 \pm 1.1$ | $68.13 \pm 1.2$ | $74.3 \pm 1.6$ | $43.7 \pm 2.3$ | $44.5 \pm 1.7$ | $47.0 \pm 1.5$ |
| **(DP-Baseline)** | | | | | | | | | |
| DP-FedAvg | $80.1 \pm 1.7$ | $85.2 \pm 1.8$ | $86.2 \pm 1.7$ | $35.17 \pm 0.8$ | $50.22 \pm 1.1$ | $74.6 \pm 1.2$ | $26.5 \pm 0.3$ | $30.7 \pm 1.4$ | $32.4 \pm 0.6$ |

the classification accuracy obtained by running the trained models on test datasets from the MNIST, CIFAR10 and CIFAR100 datasets.

**Baselines** We compare our method against the standard non-Bayesian FL algorithms and Bayesian-FL methods that build personalized models for clients. We also show results of differentially private FedAvg algorithm under similar privacy guarantee (per round $\epsilon < 0.1$) to provide perspective on the privacy. The FL baselines include - i) FedAvg, the de-facto FL learning algorithm that trains a global model, ii) FedProx, an enhancement of the FedAvg algorithm in the presence of statistical heterogeneity across clients giving a global model, iii) pFedME, which uses personalized models on each client using Monreau envelopes in loss. The Bayesian baselines include - i) pFedGP, a Gaussian process based approach that trains common deep kernels across clients and personal tree-based GPs for classification, ii) pFedBayes, which uses a variational inference-based approach for personalized FL by training personal models which are close to the aggregated global models, iii) FOLA, bayesian method using Gaussian product for model aggregation. And lastly, the DP baseline includes - i) DP-FedAvg, the FedAvg algorithm with gradient clipping and noise addition to the gradient at each client. For all the experiments, the hyper-parameters were obtained by tuning on a held-out validation dataset. We used our own implementation of the pFedBayes algorithm since the source code was not publicly available.

## 5.2 RESULTS

The performance of our method and the baselines under the non-IID data setting are reported in Table 1. Under the non-IID setting, we report the results corresponding to different dataset sizes on each client. To recall, in the small, medium, and full settings, each client has access to 50, 100, and 2500 training data points per class respectively. We observe that our method with homogeneous architectures across clients outperforms all other baselines. Moreover, when we consider the performance of our method under a heterogeneous setting by considering 30% of the total clients to be small capacity, it is evident that our method is better than the higher capacity homogeneous baselines for more complex tasks like in CIFAR-10 and CIFAR-100. On average, our method achieves about $6\%$ performance improvement over the baselines in the small and medium data settings. Figure 1 compares the performance of our method with the highest-performing baselines under model, data and statistical types of heterogeneity. Since our method can work with heterogeneous clients, we see that just by the proposed collaboration and having higher capacity clients in the FL ecosystem, the lower capacity clients are able to gain about $10\%$ increase in their performance. Also, the performance degradation of our method with a change in the number of clients with limited data resources is more graceful as compared to the baselines. In an additional experiment intended to compare the performance of the baseline methods

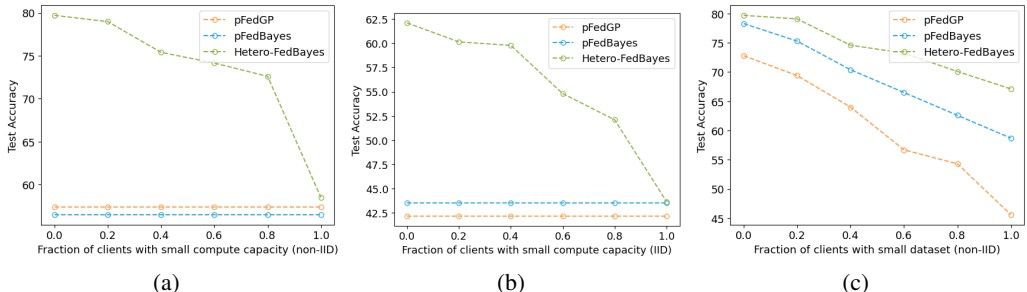

(a)                                        (b)                                        (c)

Figure 1: Performance comparison of our method with baselines under different types and varying degree of heterogeneity for CIFAR-10 dataset with 20 clients. Figure (a) is for heterogeneity in compute capacity across clients under non-IID data setting, figure (b) for compute heterogeneity under IID setting, and figure (c) for heterogeneity in data resources. When a fraction of clients in the setting have low computing resources, the baselines being homogeneous can only train smaller models on all the clients as shown by constant performance. The results show that our method is more tolerant to both model heterogeneity and data heterogeneity across clients.

with additional data, we trained the priors for baseline methods' encoders using the unlabeled data, AD, before starting their own prescribed FL procedure. We observed that the performance of the baseline methods does not change on doing this because the FL procedure that they incorporate forgets all the prior existing local knowledge at the client side. A similar result was also reported in (Sattler et al., 2021a).

The superior performance of our method could be attributed to the effective collaboration achieved via the proposed framework, wherein in each communication round, instead of aggregating clients' weight distributions and using these aggregated weights for initialization at each client, we achieve collaboration by first distilling peer knowledge in the form of the aggregated output on the AD, $\bar{\Phi}(AD)$, and then ensuring that this knowledge is successfully transferred to each client by specifying priors in the functional-space of the client model. Furthermore, the parameter in Equation 3 allows the clients the flexibility to choose the amount of global knowledge that needs to be incorporated providing flexibility on the degree of personalization.

## 6 DISCUSSION

We propose a novel method for personalized Bayesian learning in heterogeneous federated learning settings and demonstrate that our method is able to outperform the baselines under different types of heterogeneous situations, while also providing a privacy guarantee and calibrated responses. The experiments show that the method is particularly useful for clients with lower data and lower compute resources as they can benefit the most by the presence of other, more powerful clients in the ecosystem. While our method assumes the availability of a small, unlabelled auxiliary dataset at the server, it is typically a very mild requirement as such data can often be obtained from several open sources on the web. And with the recent advances in the generative AI and it's use for generating training data, it would be interesting to see if these methods can be used to readily provide application-specific alignment datasets in the future work. The privacy analysis on the method provides an intuitive and a rigorous guarantee with various tunable knobs that can be adjusted to achieve the desired privacy-utility trade-off. While the application explored in the proposed work consists of image related tasks, both the proposed framework and the privacy analysis is generic and independent of specific training algorithms, i.e., it provides a way for personalized FL for general tasks, therefore resulting in its wide applicability in various applications across data modalities. The recent use of transformer based Bayesian methods Zheng et al. (2023); Maged & Xie (2022); Tian et al. (2023) in varied applications also provides more evidence on the applicability of the proposed framework for settings where much larger neural networks are required. One limitation that originates from the Bayesian nature, and is common to all applications of Bayesian learning, is that the exact inference of the posterior distributions is infeasible and therefore variational approximation has been used for inference of the posterior distributions.

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

# Supplement for "A Bayesian Approach for Personalized Federated Learning in Heterogeneous Settings"

In this supplementary material, we first go over the preliminaries of Bayesian learning methods, followed by the pseudo-code of the algorithm used for training our framework. Then, we provide definitions and results used in the privacy analysis of our method. We show model calibration metrics and present results demonstrating our method is well-calibrated. We also discuss the details about the alignment dataset, AD, its affect on the performance and the communication and computation cost of the procedure.

## A  BAYESIAN LEARNING

Consider a learning setting where we are trying to train a neural network on a dataset $\mathcal{X}$. The aim of this setting is thus to obtain the set of weights, denoted by $\mathcal{W}$, for the corresponding neural network that best fits the data. We could also view a neural network as a model that outputs $\mathbb{P}(y|x, \mathcal{W})$ which is the distribution of the label $y$ for a given data point $x$ under the weights $\mathcal{W}$, for classification this would be the output of the softmax function. Now, the weights of the network can be learnt by Maximum Likelihood Estimation (MLE) for a given set of datapoints $\mathcal{X} = (x_i, y_i)_{i=1}^n$ by solving the following optimization problem.

$$\mathcal{W}^{MLE} = \arg\max_{\mathcal{W}} \sum_i \log P(y_i|x_i, \mathcal{W})$$

This optimization could be solved by gradient descent based methods and obtains a point estimate of the weight vector, denoted by $\mathcal{W}^{MLE}$.

The Bayesian learning methods, on the other hand, obtain a posterior distribution on the weights given the training data, $\mathbb{P}(\mathcal{W}|\mathcal{X})$ as opposed to the point estimates. The predictions for any new data point, $x$, are then obtained by taking expectation of the prediction under the posterior distribution, $y = \mathbb{E}_{w \sim \mathbb{P}(\mathcal{W}|\mathcal{X})}[\mathbb{P}(y|x, w)]$. Exact inference of the posterior distribution, however, is intractable for neural networks. Variational inference is a traditional approximation method used to obtain an approximation of the posterior weight distribution, and it has also been shown to work for neural networks (Hinton & van Camp, 1993). Specifically, variational inference tries to learn a simpler parameterized distribution $q(w|\theta)$ from a family of distributions $\mathcal{Q}$ by optimizing the parameters $\theta$ such that the new distribution $q(w|\theta^*)$ obtained for the optimal value of $\theta$ is close to the true posterior distribution $\mathbb{P}(\mathcal{W}|\mathcal{X})$. Precisely, the optimization problem looks like

$$\theta^* = \arg\min_{\theta:q(\mathcal{W}|\theta)\in\mathcal{Q}} \mathrm{KL}[q(\mathcal{W}|\theta)||\mathbb{P}(\mathcal{W}|\mathcal{X})] \tag{6}$$

$$= \arg\min_{\theta:q(\mathcal{W}|\theta)\in\mathcal{Q}} \int q(\mathcal{W}|\theta)\log\frac{q(\mathcal{W}|\theta)}{\mathbb{P}(\mathcal{W})\mathbb{P}(\mathcal{X}|\mathcal{W})} \tag{7}$$

$$= \arg\min_{\theta:q(\mathcal{W}|\theta)\in\mathcal{Q}} \mathrm{KL}[q(\mathcal{W}|\theta)||p(\mathcal{W};\psi)] - \mathbb{E}_{q(\mathcal{W}|\theta)}[\log\mathbb{P}(\mathcal{X}|\mathcal{W})] \tag{8}$$

where $p(\mathcal{W};\psi)$ signifies the prior distribution over weights $\mathcal{W}$ parameterized by $\psi$. The prior distribution is typically used to encode any previously available information about the weights of the network. The above given objective is the same objective as in Equation 1 that is used for local training in our method.

## B  ALGORITHM

The pseudo-code of the algorithm used in the FedBNN method is included in the Algorithm 1. The Algorithm 1 works in the setting when there is a server connected to $N$ clients with each client $i$ having local dataset $\mathcal{X}_i$ of size $n_i$ drawn from the local data distribution $\boldsymbol{D}_i$, and the server has an auxilliary unlabelled dataset called AD. The output of the algorithm is the set of personalized models $\Phi_i$ parameterized by $\mathcal{W}_i$ for each client $i$. All $\mathcal{W}_i$'s, instead of being point estimates, are determined by a posterior distribution $\mathbb{P}(\mathcal{W}_i|.)$ which is learnt from the data via variational inference.

As mentioned in the Section 3.2, the learning procedure first optimizes the prior parameters by minimizing Equation 4 and then learns the posterior parameters keeping the prior fixed by minimizing Equation 5.

---

**Algorithm 1** FedBNN Algorithm

---

**Input:** number of clients $N$, number of global communication rounds $T$, number of local epochs $E$, weight vector $[w_1, w_2, \ldots w_N]$, noise parameter $\gamma$
**Output:** Personalised BNNs $\{\Phi_i | i \in [1, N]\}$, parameterized by $\mathcal{W}_i \sim \mathbb{P}(\mathcal{W}_i | \mathcal{X})$
**Server Side -**
$\mathbf{X}$ = AD
**for** $t = 1$ **to** $T$ **do**
   Select a subset of clients $\mathcal{N}_t$
   **for** each selected client $i \in \mathcal{N}_t$ **do**
      $\Phi_i(\mathbf{X}) = \textbf{LocalTraining}(t, \bar{\Phi}(\mathbf{X})^{(t-1)}, \mathbf{X})$
   **end for**
   $\bar{\Phi}(\mathbf{X})^{(t)} = \sum_{j=1}^{N} w_j \Phi_j(\mathbf{X})$
**end for**
Return $\Phi_1(T), \Phi_2(T) \ldots \Phi_N(T)$

---

$\textbf{LocalTraining}(t, \bar{\Phi}(\mathbf{X})^{(t-1)}, \mathbf{X})$
Run inference on $\mathbf{X}$ to obtain $\Phi_i(\mathbf{X})$
Generate $\Phi_i^{\text{corrected}}(\mathbf{X}) = \gamma \bar{\Phi}(\mathbf{X})^{(t-1)} + (1 - \gamma)\Phi_i(\mathbf{X})$
**for** each prior epoch **do**
   Minimize CrossEntropy$(\Phi_i^{\text{corrected}}(\mathbf{X}), \bar{\Phi}(\mathbf{X})^{(t-1)})$ to obtain prior parameters $\psi$ of the BNN $\Phi_i$
**end for**
**for** each local epoch **do**
   Minimize KL$[q(\mathcal{W}_i | \theta) || p(\mathcal{W}_i; \psi^*)] - \mathbb{E}_{q(\mathcal{W}_i | \theta)}[log \mathbb{P}(\mathcal{X}_i | \mathcal{W}_i)]$ over $\{\theta : q(\mathcal{W}_i | \theta) \in \mathcal{Q}\}$ to obtain $\theta^*$
**end for**
$\mathbb{P}(\mathcal{W}_i | \mathcal{X}) \approx q(\mathcal{W}_i | \theta^*)$
Obtain $K$ Monte-carlo samples $\mathcal{W}_i^{(j)} : j \in [1, K]$ from $\mathbb{P}(\mathcal{W}_i | \mathcal{X})$
Compute $\Phi_i(\mathbf{X}) = \frac{1}{K} \sum_{j=1}^{K} \Phi_i(\mathbf{X}; \mathcal{W}_i^{(j)})$
Return $\Phi_i(\mathbf{X})$

---

## C   PRIVACY ANALYSIS

Some known results on differential privacy that are used to determine the privacy loss of our algorithm are given in this section.

A generalization of differential privacy is concentrated differential privacy(CDP). And an alternative form of concentrated differential privacy called zero-concentrated differential privacy(zCDP) was proposed to enable tighter privacy analysis (Bun & Steinke, 2016). We will also use the zCDP notion of privacy for our analysis. The relationship between standard DP and zCDP is shown below.

**Proposition C.1** (($\epsilon, \delta$)-DP and $\rho$-zCDP). *For a randomized algorithm $\mathcal{M}$ to satisfy ($\epsilon, \delta$)-DP, it is sufficient for it to satisfy $\frac{\epsilon^2}{4 log \frac{1}{\delta}}$-zCDP. And a randomized algorithm $\mathcal{M}$ that satisfies $\rho$-zCDP, also satisfies ($\epsilon', \delta$)-DP where $\epsilon' = \rho + \sqrt{4 \rho log \frac{1}{\delta}}$.*

As opposed to the notion of DP, the zCDP definition provides tighter bounds for the total privacy loss under compositions, allowing better choice of the noise parameters. The privacy loss under the serial composition and parallel composition incurred under the definition of zCDP was proved by (Yu et al., 2019) and is recalled below.

**Proposition C.2** (Sequential Composition). *Consider two randomized mechanisms, $\mathcal{M}_1$ and $\mathcal{M}_2$, if $\mathcal{M}_1$ is $\rho_1$-zCDP and $\mathcal{M}_2$ is $\rho_2$-zCDP, then their sequesntial composition given by $(\mathcal{M}_1(), \mathcal{M}_2())$ is $(\rho_1 + \rho_2)$-zCDP.*

**Proposition C.3** (Parallel Composition). *Let a mechanism $\mathcal{M}$ consists of a sequence of $k$ adaptive mechanisms, $(\mathcal{M}_1, \mathcal{M}_2, \ldots \mathcal{M}_k)$ working on a randomized partition of the $D = (D_1, D_2, \ldots D_k)$, such that each mechanism $\mathcal{M}_i$ is $\rho_i$-zCDP and $\mathcal{M}_t : \prod_{j=1}^{t-1} \mathcal{O}_j \times D_t \rightarrow O_t$, then $\mathcal{M}(D) = (\mathcal{M}_1(D_1), \mathcal{M}_2(D_1), \ldots \mathcal{M}_k(D_k))$ is $\max_i \rho_i$-zCDP.*

After computing the total privacy loss by an algorithm using the tools described above, we can determine the variance of the noise parameter $\sigma$ for a set privacy budget. The relationship of the noise variance to privacy has been shown in prior works by (Dwork & Roth, 2014; Yu et al., 2019) and is given below.

**Definition C.4** ($L_2$ Sensitivity). For any two neighboring datasets, $D$ and $D'$ that differ in at most one data point, $L_2$ sensitivity of a mechanism $\mathcal{M}$ is given by maximum change in the $L_2$ norm of the output of $\mathcal{M}$ on these two neighboring datasets

$$\Delta_2(\mathcal{M}) = \sup_{D,D'} ||\mathcal{M}(D) - \mathcal{M}(D')||_2.$$

**Proposition C.5** (Gaussian Mechanism). *Consider a mechanism $\mathcal{M}$ with $L_2$ sensitivity $\Delta$, if on a query $q$, the output of $\mathcal{M}$ is given as $\mathcal{M}(x) = q(x) + \mathcal{N}(0, \sigma^2)$, then $\mathcal{M}$ is $\frac{\Delta^2}{2\sigma^2}$-zCDP.*

## D  CALIBRATION

Model calibration is a way to determine how well the model's predicted probability estimates the model's true likelihood for that prediction. Well-calibrated models are much more important when the model decision is used in critical applications like health, legal etc. because in those cases managing risks and taking calculated actions require a confidence guarantee as well. Visual tools such as reliability diagrams are often used to determine if a model is calibrated or not. In a reliability diagram, model's accuracy on the samples is plotted against the confidence. A perfectly calibrated model results in an identity relationship. Other numerical metrics that could be used to measure model calibration include Expected Calibration Error (ECE) and Maximum Calibration Error (MCE). ECE measures the expected difference between model confidence and model accuracy whereas MCE measures the maximum deviation between the accuracy and the confidence. The definitions and empirical formulas used for calculating ECE and MCE are as given below.

$$\text{ECE} = \mathbb{E}_{\hat{P}}[\mathbb{P}(\hat{Y} = Y | \hat{P} = p) - p],$$

$$\text{MCE} = \max_{p \in [0,1]} |\mathbb{P}(\hat{Y} = Y | \hat{P} = p) - p|.$$

Empirically,

$$\text{ECE} = \sum_{i=1}^{M} \frac{|B_i|}{n} |\text{accuracy}(B_i) - \text{confidence}(B_i)|,$$

$$\text{MCE} = \max_{i \in [1,M]} |\text{accuracy}(B_i) - \text{confidence}(B_i)|,$$

where $B_i$ is a bin with set of indices whose prediction confidence according to the model falls into the range $\left(\frac{i-1}{M}, \frac{i}{M}\right)$. Figure 2 shows the reliability diagram along with the ECE and MCE scores for our method measured on MNIST and CIFAR-10 dataset in the non-IID data setting.

## E  ALIGNMENT DATASET (AD)

In FedBNN, the alignment dataset (AD) is used to achieve collaboration across clients. Since the only assumption on AD is for it to be of the same domain as the target application, there is no practical constraint on obtaining the AD in real-world settings. In many cases it could be obtained from web, for example images from common datasets in Huggingface, texts from Wikipedia, Reddit etc. The use of AD is not different from how several other methods use an additional dataset for augmentation. The effect of size of AD on the performance of models is demonstrated in Figure 3 for CIFAR-10 dataset in the small data and non-IID setting. In that figure, we observe that when the size of AD is small the performance of the model is low but as the size of AD increases the performance increases up to a point and becomes constant afterwards. The number of data points in AD that are required to achieve good improvement in the model performance is small and practical.

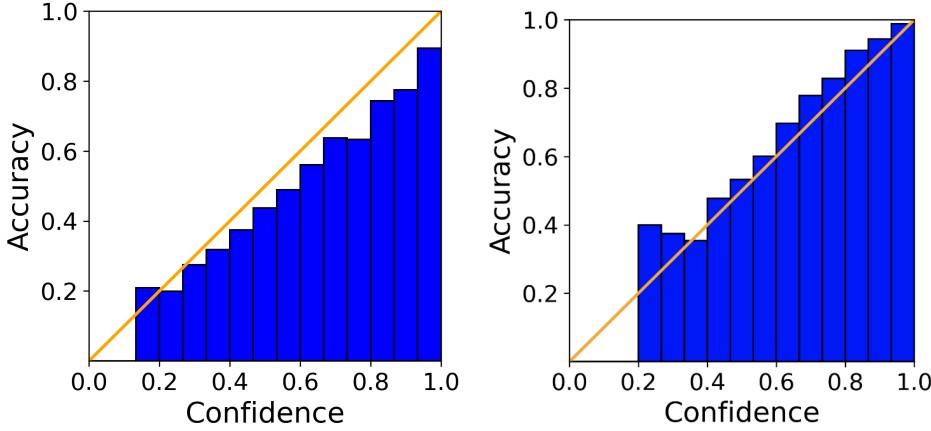

(a) **Dataset**: CIFAR-10, **ECE**: 0.070, **MCE**: 0.134  (b) **Dataset**: MNIST, **ECE**: 0.032, **MCE**: 0.156

Figure 2: Reliability diagrams and scores showing model calibration. Figure (a) is for the results corresponding to the CIFAR-10 dataset and Figure (b) for MNIST dataset.

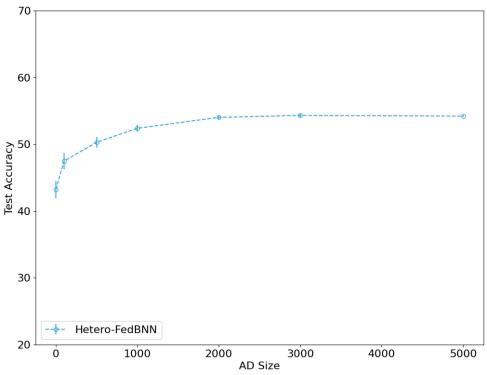

Figure 3: Ablation study comparing the affect of AD size on the performance. The included results are for CIFAR-10 dataset in the small data setting with non-IID partitions and heterogeneous clients.

We also vary the distribution of the AD being used and test the final performance of the models and report it in Table 2. We run these experiments on 20 clients for CIFAR-10 dataset where each client had access to only 5 of the 10 classes and each client belonged to the medium data setting. For the first experiment, we use a held-out dataset from the CIFAR-10 data as AD but vary the composition of the dataset by changing the distribution of the classes present in the AD, for example, CIFAR10(10) is composed of all 10 classes present in the CIFAR-10 dataset but CIFAR10(2) is composed of only 2 out of the 10 classes present in the AD and likewise. We also test the performance of our method when a significantly different dataset SVHN consisting of the colored house number images is used. Table 2 suggests that the performance of the method even with different datasets as AD *always* improves and that the gain between local training and the proposed procedure is better highlighted in the heterogeneous architecture settings, since there local client capacities and model architectures differ significantly and clients are able to utilize the peer knowledge to learn better models locally. We observed that even for different and dissimilar data distributions in AD, it is possible to obtain a value for the parameter $\gamma$ such that the final performance of the local client model with collaboration is better than the model independently trained locally on the client. The parameter $\gamma$ controls the amount of global knowledge to be incorporated on each client and helps in regularization of the client models which enables them to generalize better and perform better on the test data.

Table 2: Effect of varying distribution of AD on the clients' performance for the non-IID seeting with CIFAR-10 dataset and 20 clients where each client has data for the 5 different classes.

| Architecture Setting | Local Training | CIFAR10(10) | CIFAR10(8) | CIFAR10(5) | CIFAR10(2) | SVHN |
|---|---|---|---|---|---|---|
| Homogeneous Architectures | $64.3 \pm 0.36$ | $72.7 \pm 0.15$ | $69.7 \pm 0.28$ | $68.8 \pm 0.97$ | $67.2 \pm 1.5$ | $70.1 \pm 0.18$ |
| Heterogeneous Architectures | $61.2 \pm 0.17$ | $71.6 \pm 0.93$ | $68.4 \pm 0.80$ | $68.8 \pm 1.4$ | $68.1 \pm 1.9$ | $69.3 \pm 0.8$ |

## F  COMMUNICATION AND COMPUTATION EFFICIENCY

**Communication Cost**    In FedBNN, each global communication round requires that the server sends the alignment dataset to all the clients and the clients upload the outputs of their respective models on the common dataset AD. Since AD is a publicly available dataset, AD could be transmitted to the clients by specifying the source and the indices, and does not really needs to be communicated across the channel. The client output on AD, on the other hand, depends on the number of instances in AD, let's call it $K$, therefore, the total communication cost in each round of our method is $O(K)$. As shown in Figure 3, having $K = 2000$ gives a good performance. The communication cost between the clients and the server, thus, is also invariant of the number of model parameters which tends to run in millions. This allows our method to be much more communication efficient as compared to the conventional FL algorithms that transmit model parameters in each communication round, making it practically more useful.

**Computation Cost**    Similarly, the computation cost of a FL procedure involves the costs incurred in local training at the individual clients and the cost of aggregation at the server, both of which are discussed below.

- **Server-side computation cost** The server side computation cost arises from the need to aggregate knowledge obtained from individual clients. In the state-of-the-art bayesian FL algorithms, the server aggregates posterior distributions for each weight parameter in the neural network obtained from various clients. The number of such weight parameters typically run in millions. In our method we do not aggregate the parameter distributions but achieve collaboration by aggregating the client outputs on the AD (with size  2000), thus the server side computation cost in our method is many orders of magnitude lower than the conventional methods and does not depend on the number of model parameters. This makes our method much more efficient and scalable than existing federated bayesian solutions.

- **Client-side computation cost** The client-side computation cost is mostly determined by the cost of training a Bayesian Neural Network at the client side, which in turn depends on the type of inference procedure used for obtaining the posterior distribution over weights for each parameter of the neural network. In the proposed work, the method used for inference is Bayes by Backprop which uses the gradient computations similar to backpropagation(BP) algorithm to obtain the posterior distributions where the posterior distributions are characterized by the mean and standard deviation. A re-parameterization trick is used to compute the mean and std of the distributions from the backpropagated gradients. Thus the cost for obtaining the posterior distributions is similar to the cost of backpropagation. Moreover, since the method only uses gradient updates, the optimizations used for SGD like asynchronous SGD etc. could be readily used for obtaining the posteriors. An unrelated but similar algorithm in (Hernández-Lobato & Adams, 2015) does probabilistic backpropagation to train BNNs and shows that the average run time of probabilistic BP is not higher than that of BP.

To summarize, the communication cost and the server-side computation cost of the proposed method is orders of magnitude lower than that of the other Bayesian baseline methods. On the other hand, the client-side computation cost is determined by the inference procedure used to obtain the posterior distributions and for which Bayes by Backprop provides an efficient mechanism. Several works in the recent past have discussed the use of related Bayesian inference based methods for training uncertainty-aware transformers (Zheng et al., 2023; Tian et al., 2023; Maged & Xie, 2022) proving that Bayesian methods are not limited to use in simpler models. And therefore, our framework can also be extended to apply in settings where much larger neural networks are required.

Table 3: Performance comparison as a function of the privacy guarantee.

| Privacy ($\epsilon$) per round | Test Accuracy |
|---|---|
| $\approx 1$ | 75.5 % |
| $\approx 0.1$ | 71.3 % |
| $\approx 0.01$ | 68.6 % |
| $\approx 0.001$ | 62.2 % |
| $\approx 0.0001$ | 59.6 % |

Table 4: Test accuracy comparison with more number of clients (500) in the setting.

| Method | Test Accuracy |
|---|---|
| pFedGP | $53.2 \pm 0.4$ |
| pFedBayes | $52.9 \pm 0.8$ |
| Ours(Homo) | $56.1 \pm 0.3$ |
| Ours(Hetero) | $54.7 \pm 1.0$ |

## G  ADDITIONAL EXPERIMENTS

**Privacy vs Performance**   Since the amount of noise required to be added to the client's outputs via the Gaussian Mechanism is directly proportional to the guaranteed privacy, we test the affect of the privacy guarantee on the performance of the proposed framework by comparing the performance of the method with varying $\epsilon$ and $\delta = 10^{-4}$. The results are reported in Table 3. We observe that, as expected, when we reduce the amount of privacy loss in each iteration by adding more noise to the clients' outputs going to the server, the performance of the method drops. However the drop in performance in all the cases is not drastic as the clients can tune the level of personalization or global knowledge required by appropriately setting the parameter $\gamma$ in Equation 3.

**More clients**   To test the performance of the proposed method when a large number of clients are involved in the setup, we did additional experiments with 500 clients and non-IID setting with 5 classes per client in the medium data setting on the CIFAR-10 dataset where in each communication round only 10% of the clients are selected for participation and $\gamma = 0.7$. The obtained results at the end of $200^{th}$ communication round are reported in Table 4. We observe that the homogeneous version of our method is better than the baselines by a significant margin and the heterogeneous version is slightly better than the baselines.

