# OpenReview forum: "A Bayesian Approach for Personalized Federated Learning in Heterogeneous Settings"
_ICLR.cc/2024/Conference — Submitted to ICLR 2024_

### Official Review · Reviewer_j79p · 2023-10-19

**Soundness:** 3 good
**Presentation:** 2 fair
**Contribution:** 3 good
**Rating:** 6
**Confidence:** 3

**Summary:**

This paper proposes a personalized federated learning framework based on Bayes Learning methods to tackle the heterogeneity. Specifically, this paper proposes FedBNN, which utilizes a globally shared alignment dataset to collect the information of the models on all the clients and find a better prior distribution for the local model. With the optimized prior, a personalized posterior is obtained by the local variational inference. DP-based methods can be applied to FedBNN to guarantee privacy.

**Strengths:**

This paper proposes a novel Bayes Learning framework FedBNN which attains significantly better performance compared with previous baselines under the heterogeneous settings. More specifically, the strengths of this paper lie in the following aspects:

1.	FedBNN successfully addresses both systematic and statistical heterogeneity in federated learning by adopting functional sharing instead of model sharing. This makes the framework more practical and flexible.

2.	A DP-based method is provided to guarantee the privacy of FedBNN with rigorous proof.

3.	Sufficient experiments show that FedBNN outperforms previous baselines under highly heterogeneous settings while maintaining high privacy.

**Weaknesses:**

While the proposed method is shown to be promising, some clarity issues should be addressed to improve this paper. More specifically,

1.	More details of Bayes Learning could be introduced with mathematical definitions. For a reader who is not so familiar with Bayes Learning, it would be helpful to include a basic framework of Bayes Learning. For example, the optimization objective and training procedure of standard Bayes Learning, as well as the definition of $q(W_i|\theta)$ and $p(W_i;\psi)$.

2.	It appears to me that the significance of FedBNN mainly comes from the utilization of the public alignment dataset instead of Bayes Learning. It would be better to include a baseline that trains the model $W_i$ directly instead of using Bayes Learning. Namely, the public alignment dataset is used for knowledge distillation at the beginning of each communication round to initialize $W_i$, and then $W_i$ is trained with private local data.

**Questions:**

1.	What is the form of $p(W_i;\psi)$? Is it also a Gaussian Distribution parameterized by $\psi$?

2.	Are the local datasets of different clients overlapped? For example, in the full setting of CIFAR10, each client has 5 classes and thus 12500 samples, which requires 250000 samples in total without any overlapping. This is not a standard setting in Federated Learning.

---

> ### Author Response · Authors · 2023-11-20
> **Response to Reviewer j79p**
>
> We sincerely thank the reviewer for reviews and the positive feedback on our paper. We are encouraged to see that the reviewer finds our method promising with sufficient experiments outperforming the baselines and the DP variant and privacy guarantee useful. We respond to all of reviewer's questions and comments below.
>
> -- *"More details of Bayes Learning could be introduced with mathematical definitions."*
>
> We thank the reviewer for suggesting this. We have included a section in the Appendix of the revised paper that introduces Bayesian Learning and variational inference for Bayesian learning along with the definitions of $q(\mathcal{W}_i|\theta)$, $p(\mathcal{W}; \psi)$ and other terms used in the context of Bayesian learning in the paper.
>
> -- *"It would be better to include a baseline that trains the model $\mathcal{W}_i$ directly instead of using Bayes Learning"*
>
> We again thank the reviewer for suggesting this. We have included this baseline in the revised paper under the name "KD based collaboration" (Knowledge-distillation based collaboration) under non-Bayesian baselines.
>
> -- *"What is the form of $p(\mathcal{W}; \psi)$? Is it also a Gaussian Distribution parameterized by $\psi$?"*
>
> $p(\mathcal{W}; \psi)$ denotes the prior distribution and is indeed a Gaussian distribution parameterized by $\psi$, each parameter of the network is assumed to have a Gaussian prior distribution. Specifically, for a BNN parameterized by a set of weights $\mathcal{W_i}$, $\psi = \{(\mu_{i,\alpha}^{(p)\}, \sigma^{(p)}_{i,\alpha}), \alpha \in [1, \dots, |\mathcal{W}_i|]}$.
>
> -- *"Are the local datasets of different clients overlapped? For example, in the full setting of CIFAR10, each client has 5 classes and thus 12500 samples, which requires 250000 samples in total without any overlapping. This is not a standard setting in Federated Learning."*
>
> The local datasets are not overlapped. By full we mean the number of data points that get assigned to a specific client while simulating the FL setting, as opposed to the small and medium data settings where we restrict the number of data points that can be present per class on each client. In the CIFAR-10 example, if 4 clients have access to class 1, class 1 points will be partitioned randomly to all 4 of those clients with each client now having 125 data points for class 1. And if each client has access to 5 classes then 125x5 = 625 data points per client.

---

> > ### Author Response · Authors · 2023-11-22
> > **Regarding our response**
> >
> > Dear reviewer j79p,
> >
> > We sincerely appreciate your feedback and have carefully addressed the concerns in our response, and made modifications to the revised paper. If you have any further questions or comments, kindly let us know, we would be happy to address them as well. Thank you!

---

### Official Review · Reviewer_wAJ2 · 2023-10-26

**Soundness:** 2 fair
**Presentation:** 2 fair
**Contribution:** 1 poor
**Rating:** 3
**Confidence:** 3

**Summary:**

This paper studies personalized federated learning algorithms under differential privacy constraints.  Consider a set of clients connected to a central server, where each client has a local dataset. The goal is to design individual models for each client while preserving privacy of the clients' data with the help of the Alignment Dataset (AD) available on the server side. DP federated learning algorithm based on the Bayesian approach has been proposed. At each round, each sampled client runs a local optimizer to update her own model, and then the client privately reports the output of the local NN on the alignment dataset (AD). The server aggregates the clients' reports and sends the aggregation to the sampled clients in the next round. The aggregated output obtained from the server represents the collaborative information between clients and is used to improve learning the individual models.

**Strengths:**

I think the new idea in this paper is the following. Instead of sharing the local models $\mathcal{W}_i$, each client shares how the local model $\mathcal{W}_i$ classifies the shared dataset (Alignment Dataset(AD)).  Then, each client improves her local training by distilling information from the aggregated outputs.  However, it is not clear to me why this might be a useful idea in personalized learning.

**Weaknesses:**

In my opinion, there is a lack of explanation and justification for the theoretical model. For example, what is the assumption on distributions $\mathbb{P}[\mathcal{X}]$, and $\mathbb{P}[\mathcal{X}|\mathcal{W}_i]$.  Also, there are missing lots of details of the main algorithm and a lack of theoretical analysis of the proposed scheme. The existence of a public dataset might be a strong assumption. Furthermore, it is not easy to share this public dataset with all clients in federated learning. Please, read my questions below.

**Questions:**

- What is the assumption that connects the local model $\mathcal{W}_i$ and the local dataset $\mathcal{X}_i$? It is supposed to be a probabilistic model of how the data is generated for a given local model $\mathcal{W}_i$.

- In Section 3.2 (local setting): Are the prior distributions means $\lbrace\mu_i\rbrace$ and variances $\lbrace\sigma_i\rbrace$ are unknown to the clients? Why is the assumption that the models are generated from Gaussian distribution? Which step in the algorithm this assumption is used for?

- Why the shared data $\Phi_i(AD)$ is a useful information? What if the alignment dataset (AD) has a distribution that is completely different from the distribution of the local datasets $\lbrace \mathcal{X}_i\rbrace$?

- Why the assumption that the distribution $\mathbb{P}[\mathcal{W}_i|\mathcal{X}]$ lies in the family of Gaussian distributions? Even if $\mathbb{P}[\mathcal{W}_i]$ has a prior Gaussian distribution, that doesn't mean that $\mathbb{P}[\mathcal{W}_i|\mathcal{X}]$ is Gaussian.

- What is $p(\mathcal{W}_i;\psi)$ defined before Eqn (1) refers to and how this parameter is connected to the true distribution $\mathbb{P}[\mathcal{W}_i|\mathcal{X}]$

- Could you please explain in more detail how we get (2) from (1)?

- Please explain in Theorem 4.2 what is the value of the variance $\sigma_g^2$ to achieve $\left(\varepsilon,\delta\right)$-DP.

- The $L_2$ sensitivity of the model is not clear to me. As far as I understand, $\Phi_i(AD;\mathcal{W}_i)$ denotes the output of the NN on the alignment dataset (AD). Thus,  $\Phi_i(AD;\mathcal{W}_i)$ has the same size as the number of samples in the dataset AD right? Thus, the sensitivity is supposed to be a function of the size of the AD. Please, explain as I might be missing something.

---

> ### Author Response · Authors · 2023-11-17
> **Response to Reviewer wAJ2 (1/2)**
>
> We thank the reviewer for the feedback. We address the concerns raised by the reviewer as follows.
>
> We first clarify the overall contribution of the paper. The paper addresses the problem of enabling efficient federated learning of Bayesian networks under heterogeneous settings. Bayesian learning methods are useful when posterior distribution over the individual parameters (weights) of the neural network are desired, when we have very limited data or in applications where determining the uncertainty in the prediction is important. As mentioned in the introduction section of the paper, typical Bayesian-FL approaches that require computing the local posteriors and aggregating them to create global posteriors is computationally prohibitive for large models such as neural networks. Furthermore, when the local models are not identical, aggregating the local posteriors is not straightforward. In this work, we first propose a new method of initializing the prior distribution over weights of the networks via the output space, and then to enable collaboration across clients we obtain the collective knowledge from the peer clients on an auxiliary dataset called AD, i.e., the collective knowledge is obtained and distilled to each client along with its local training on the local data. The collective knowledge distillation helps the local client to learn from other peer clients in the setup since now other clients can act as a teacher for this local network without actually sharing their data or explicitly sharing millions of model parameters.
>
> Knowledge distillation for achieving collaboration in FL settings has also been successfully used previously for training heterogeneous models in non-Bayesian settings [1,2,3]. But none of the prior works have studied it in the context of Bayesian learning.
>
> -- *"In my opinion, there is a lack of explanation and justification for the theoretical model. For example, what is the assumption on distributions $\mathbb{P}[\mathcal{X}]$, and $\mathbb{P}[\mathcal{X}|\mathcal{W}_i]$. What is the assumption that connects the local model $\mathcal{W}_i$ and the local dataset $\mathcal{X}_i$? It is supposed to be a probabilistic model of how the data is generated for a given local model $\mathcal{W}_i$? Are the prior distributions means $\{\mu_i\}$ and variances $\{\sigma_i\}$
>  are unknown to the clients? Why is the assumption that the models are generated from Gaussian distribution? Which step in the algorithm this assumption is used for?"*
>
> There is no assumption on the data generative process or $\mathbb{P}(\mathcal{X})$ or  $\mathbb{P} ( \mathcal{X} | \mathcal{W_i} )$. Rather, like in the standard Bayesian learning, we are trying to infer the parameters $\mathcal{W_i}$ by not obtaining the point estimates of each weight, but obtaining a distribution over the weights $\mathbb{P}(\mathcal{W_i} | \mathcal{X_i})$ such that the distribution of weights best explains the data. Each local weight vector, $\mathcal{W_i}$ is composed of many weights $w_{i,\alpha }$ for $\alpha \in [1, \dots, |\mathcal{W_i}|]$ corresponding to the number of parameters in the neural network, and the goal of the Bayesian learning process is to infer the distribution of each of these parameters given the data, i.e., $\mathbb{P}(w_{i,\alpha} | \mathcal{X_i})$. To simplify the learning process slightly, we assume that each of $w_{i,\alpha}$ is a Gaussian distribution, $\mathcal{N}(\mu_{i,\alpha}, \sigma^2_{i,\alpha})$, with unknowns $\mu_{i,\alpha}$ and $\sigma^2_{i,\alpha}$. This is a standard assumption in the Bayesian learning methods and is also made in all the Bayesian baselines. Therefore, now the learnable parameters for each local neural network include the mean $\mu_{i,\alpha}$ and variance $\sigma^2_{i,\alpha}$ of each of the weight and together they characterize the local posterior distribution of weights, $\mathbb{P}[\mathcal{W_i}|\mathcal{X_i}]$, for each local client. The form of the prior distribution is also considered similar with its parameters denoted as $\psi = \{ (\mu_{i,\alpha}^p, \sigma^{2,p}_{i,\alpha}), \alpha \in [1, \dots, |\mathcal{W}_i|] \} $ and the prior parameters are learnt by the knowledge distillation process. This assumption is used in solving the learning objective in Equation 5 of the paper. Additional details about Bayesian based learning and variational inference procedure for Bayesian learning are now also included in the Appendix of the updated paper.
>
> [1] Li et al., FedMD: Heterogenous Federated Learning via Model Distillation
>
> [2] Sattler et al., FedAUX: Leveraging Unlabeled Auxiliary Data in Federated Learning
>
> [3] Goodfellow et al., Semi-supervised Knowledge Transfer for Deep Learning from Private Training Data

---

> ### Author Response · Authors · 2023-11-18
> **Response to Reviewer wAJ2 (2/2)**
>
> We address the other comments from the reviewer as follows:
>
> -- *"Why the assumption that the distribution $\mathbb{P}(\mathcal{W}_i|\mathcal{X})$ lies in the family of Gaussian distributions? Even if has a prior Gaussian distribution, that doesn't mean that $\mathbb{P}(\mathcal{W}_i|\mathcal{X}_i)$ is Gaussian."*
>
> The Gaussian assumption on the parameters of a Bayesian Neural Network is a standard assumption since it allows comprehensive understanding of the theory, simplifies the learning process and analysis of the obtained parameter distributions. When a large number of model parameters are involved. It has been used in all the considered Bayesian baselines as well as other popular methods of Bayesian Learning.
>
> -- *"What is $p(\mathcal{W}_i; \psi)$ defined before Eqn (1) refers to and how this parameter is connected to the true distribution $\mathbb{P}[\mathcal{W}_i|\mathcal{X}]$."*
>
> $p(\mathcal{W}_i; \psi)$ denotes the prior distribution over weights $\mathcal{W}_i$ which is typically used to contain any prior knowledge about the weights of the BNN and is parameterized by $\psi$. The posterior distribution $\mathbb{P}[\mathcal{W}_i|\mathcal{X}]$ is connected to the prior distribution via the Bayes' rule where posterior $\propto$ prior $\times$ likelihood, where likelihood depends on the data in the learning process. The two terms of the Equation 2 in the paper indeed consist of the the prior part in the first term and the data likelihood part in the second term.
>
> -- *"Could you please explain in more detail how we get (2) from (1)?"*
>
> By applying the definition of KL divergence and distributing the terms over log, we have the following simplification.
> $$
> \begin{align}
>  \theta &= \underset{\theta : q(\mathcal{W}| \theta) \in \mathcal{Q}}{\operatorname{argmin}}  \text{KL}[q(\mathcal{W} | \theta) || (\mathcal{W} | \mathcal{X})] \\
>  &= \underset{\theta : q(\mathcal{W}| \theta) \in \mathcal{Q}}{\operatorname{argmin}} \text{log} \dfrac{q(\mathcal{W} | \theta)}{\mathbb{P}(\mathcal{W}) \mathbb{P}(\mathcal{X}|\mathcal{W})} \\
>  &= \underset{\theta : q(\mathcal{W}| \theta) \in \mathcal{Q}}{\operatorname{argmin}} \text{KL}[q(\mathcal{W} | \theta) || p(\mathcal{W}; \psi)] - \mathbb{E}_{q(\mathcal{W} | \theta)}[\text{log} \mathbb{P}(\mathcal{X}| \mathcal{W}]
> \end{align}
> $$
> we apologise that due to lack of space we are including this simplification in Appendix under the Bayesian learning section.
>
> -- *"Please explain in Theorem 4.2 what is the value of the variance $\sigma_g^2$ to achieve $(\epsilon, \delta)$-DP."*
>
> The value of $\sigma$ required to achieve $(\epsilon, \delta)$-DP depends on how tight the privacy is, the tighter the privacy the higher the standard deviation in the noise. For the given experiment, we constrain the $\epsilon$ to be a single digit and $\delta = 10^{-4}$ and for this we need a rather higher standard deviation of about $\approx 100$.
>
> -- *"The $L_2$ sensitivity of the model is not clear to me. As far as I understand, $\Phi(AD)$ denotes the output of the NN on the alignment dataset (AD). Thus, $\Phi(AD)$ has the same size as the number of samples in the dataset AD right? Thus, the sensitivity is supposed to be a function of the size of the AD. Please, explain as I might be missing something."*
>
> The $L_2$ sensitivity of a model is defined with respect to two neighboring datasets $D$ and $D'$ that differ in at most one data point. As defined in Definition B.4, $L_2$ sensitivity measures the maximum change in output of the model when this model is applied to two neighboring datasets and since the $\Phi(AD)$ denotes the logit output of the NN, maximum difference between $\Phi(AD)$ and $\Phi(AD')$ where $AD$ and $AD'$ are two neighboring datasets is bound by 2.
>
> -- *Regarding the public dataset.*
>
> Several applications can make use of publicly available open source datasets like they are used for wide pre-training of other large scale models, for example, learning tasks related to image classification can use image datasets like MNIST or CIFAR for training which are publicly available, text based tasks can use wikipedia, similarly acoustic applications can use audio datasets provided by Huggingface, etc. As shown in Figure 3 and table 2 in Appendix D of the original paper, the size of the dataset need not be large and even the different distributions of the datasets help in learning. In the discussion section of the original paper, we also mention how the recent advancements in use of generative AI for generating training data for various modalities is a promising future direction for this work.

---

> ### Author Response · Authors · 2023-11-21
> **Regarding our response**
>
> Dear reviewer wAJ2,
>
> We appreciate your feedback and have carefully addressed the concerns in our response, and made modifications to the revised paper. If you have any further questions or comments, kindly let us know, we would be happy to address them as well.

---

### Official Review · Reviewer_aEw6 · 2023-10-31

**Soundness:** 2 fair
**Presentation:** 2 fair
**Contribution:** 3 good
**Rating:** 5
**Confidence:** 4

**Summary:**

This paper proposes a Bayesian framework for personalized federated learning, allowing different clients to train different architectures. This approach assumes an auxiliary public dataset (AD), which is used to initialize the local priors and share information (aggregated model output logits) between client and server. Each client has its own Bayesian neural network and initializes its prior by optimizing its model outputs to be close to the global model outputs. The local optimization then approximates the local posterior via variational inference. Finally, the model outputs from the new local model are shared to the server. To protect the privacy, the authors proposed to add local DP to the shared model outputs with Gaussian mechanism. This approach is evaluated on several benchmark datasets with different heterogeneous settings and is shown to be superior to the previous works.

**Strengths:**

- This paper is well-motivated as Federated learning in practice can experience extreme heterogeneous settings and one global model architecture will likely not fit all clients.
- The personalization method is novel where only the outputs instead of any weight parameters will be shared, which could be more communication-efficient for larger models. The local DP also provides a strong privacy guarantee. And the results from Table 1 indicated that the proposed method is superior to the others, though it does have an extra assumption that a good AD exists.

**Weaknesses:**

- The proposed method seems to heavily depend on how good AD is. Indeed, for common image and text tasks, it might be easy to find such a public dataset. But for more sensitive tasks on devices, such a public dataset might not exist.
- Scale of experiments is small, where the tasks such as MNIST or CIFAR10 are relatively simple. It is hard to know whether the method can generalize to larger models or harder tasks by just sharing the model outputs.
- Local DP noise results with such a small epsilon seems to be unreasonably good, as they are nearly all better than the non-DP baseline for CIFAR. From Theorem 2, with $(\epsilon, \delta)=(5, 10^{-4}), E=200, K=2000$, then $\rho\approx 1.7 * 10^{-6}$, the noise standard deviation is about 767 which is much larger than the output scale. It would be great if the authors can explain how local prior optimization is not impacted by DP noise and outperform the non-DP baselines.
- Also since the authors considered a public dataset is available, then the DP baseline should also be those with such assumptions, such as [1].
- Minor: presentation of the hierarchy in Algorithm 1 can be improved.

References

[1] Li, Tian, et al. "Private adaptive optimization with side information." International Conference on Machine Learning. PMLR, 2022.

**Questions:**

- Why not consider central DP (where the noise is added once on the aggregated output), which is a more common practice in federated learning?
- In the appendix, why does SVHN (an OOD dataset) as AD give better results than CIFAR10 as AD?
- Why is the sensitivity of the model outputs bounded by 2? How did this bound get enforced?
- There seems to be a simpler non-Bayesian alternative: clients can use the aggregated outputs and train a smaller model on AD using distillation, then fine-tune the distilled model on their private dataset. How would this baseline compare to the Bayesian approach?

---

> ### Author Response · Authors · 2023-11-20
> **Response to Reviewer aEw6 (1/2)**
>
> We sincerely thank the reviewer for the valuable feedback. It is encouraging for us to see that the reviewer finds our problem to be well-motivated and the method to be novel with superior experimental results.
>
> We address each of the reviewer's questions and concerns below.
>
> -- *"Why not consider central DP (where the noise is added once on the aggregated output), which is a more common practice in federated learning?"*
>
> The learning procedure in federated learning involves the clients' sharing their model parameters or outputs to the server, server aggregating the clients' outputs and then broadcasting the aggregated outputs to all the other clients for next rounds of training. If we add noise once to the aggregated output, it would mean that noise is getting added at the server after the aggregation. While this prevents the leakage of the data from one client to another client, this does not prevent loss of privacy due to transfer of information from the client to the server. The private version of the algorithm proposed in the paper and the privacy analysis are designed to provide stronger privacy by also preventing the data leakage from the client to the server. This is particularly more important in settings where server is honest but curious or when the server is easier to be compromised.
>
> -- *"In the appendix, why does SVHN (an OOD dataset) as AD give better results than CIFAR10 as AD?"*
>
> With respect to the distribution of AD, the best results for CIFAR-10 classification are seen when AD is composed of a held-out set from all 10 classes of CIFAR-10 denoted as CIFAR10(10) which is as expected. Then, as the composition of AD is changed from 10 classes to random 8, 5 and 2 classes of CIFAR-10 (denoted as CIFAR10(8), CIFAR10(5) and CIFAR10(2) respectively) the performance keeps on decreasing. We see that the performance of the same task with SVHN as AD is only strictly better than CIFAR10(2) as AD. Now the effect of the AD on the local client's performance is via $\Phi_i^{corrected} = \gamma \bar{\Phi}(AD) + (1- \gamma)\Phi_i(AD)$. With appropriately set $\gamma$, the AD also helps achieve regularization for the local client model such that the local models do not overfit to the relatively smaller local datasets. We believe that  the SVHN dataset as AD works better than the CIFAR10(2) because it provides more variability in the data distribution. A similar observation was also recorded in FEDBE [2] which also uses additional unlabelled data for knowledge distillation, which noted that the out-of-domain unlabelled data for distillation can perform even better.
>
> [2] Chen et al., FEDBE: MAKING BAYESIAN MODEL ENSEMBLE APPLICABLE TO FEDERATED LEARNING, ICLR 2021.
>
> -- *"Why is the sensitivity of the model outputs bounded by 2? How did this bound get enforced?"*
>
> The $L_2$ sensitivity of a model is defined with respect to two neighboring datasets $D$ and $D'$ that differ in at most one data point. As defined in Definition B.4, $L_2$ sensitivity measures the maximum change in output of the model when this model is applied to two neighboring datasets and since the $\Phi(AD)$ denotes the logit output(output of softmax function) of the NN, maximum difference between $\Phi(AD)$ and $\Phi(AD')$ where $AD$ and $AD'$ are two neighboring datasets is bound by 2.
>
> -- *"There seems to be a simpler non-Bayesian alternative: clients can use the aggregated outputs and train a smaller model on AD using distillation, then fine-tune the distilled model on their private dataset. How would this baseline compare to the Bayesian approach?"*
>
> We thank the reviewer for suggesting this. We have added this baseline in the revised paper under the name "KD based collaboration" (Knowledge-distillation based collaboration).
>
> -- *"The proposed method seems to heavily depend on how good AD is. Indeed, for common image and text tasks, it might be easy to find such a public dataset. But for more sensitive tasks on devices, such a public dataset might not exist."*
>
> The additional ablation studies done on the distribution and size requirement of the auxiliary alignment dataset, AD, shown in the Appendix D of the original paper tell us that the size of the AD need not be large and that the distribution of AD also need not be aligned with the clients' local data distributions. The only assumption on AD is to be of the same modality as the clients' data and since several applications including image, text, acoustics, autonomous driving, healthcare etc., can make use of publicly available open source datasets like they are already used for wide pre-training of other large scale models today, most tasks do have some related public open source datasets. We also mention in paper how the recent advancements in use of generative AI for generating training data for various modalities is a promising future direction for this work where we can use the synthetic generated data as AD. [3,4] also show real settings where public datasets are available.

---

> ### Author Response · Authors · 2023-11-20
> **Response to Reviewer aEw6 (2/2)**
>
> We address the other comments from the reviewer as follows:
>
>
> -- *"Scale of experiments is small, where the tasks such as MNIST or CIFAR10 are relatively simple. It is hard to know whether the method can generalize to larger models or harder tasks by just sharing the model outputs."*
>
> The choice of the datasets and experiment settings is primarily motivated by the experiment settings used in the baseline method papers with the goal of showing the efficiency of the approach. Apart from MNIST and CIFAR10, we also show compare performance on CIFAR100 and see that the gap in the performance holds showing that our method can tackle more complex tasks. We also have experiments with more number of clients in the Appendix to show that our method can handle overall more complex settings as well.
>
> -- *"Also since the authors considered a public dataset is available, then the DP baseline should also be those with such assumptions, such as [1]."*
>
> We thank the reviewer for suggesting this. While the baseline is only applicable to homogeneous architecture clients, we performed initial experiments comparing our method and report initial results below. As the available code for the suggested baseline [1] readily supports experiments on MNIST, we report initial results with MNIST dataset but we will report further results in the camera ready version.
>
> |Method | small data | medium data | large data |
> |----------|-------------|-------------|-------------|
> AdaDPS [1] | 86.7 | 89.6 | 91.9 |
> Our method (DP variant) | 89.8 | 90.2 | 91.4 |
>
> [1] Li, Tian, et al. "Private adaptive optimization with side information." International Conference on Machine Learning. PMLR, 2022.
>
> -- *Regarding the impact of noise*
>
> After adding the noise to the model output on each client, the noisy versions are communicated to the server where an aggregation is performed on the noisy versions to obtain the aggregated output, $\bar{\Phi}(AD)$, which is then broadcast to all the clients for next round of training. Before using the aggregated output directly, the clients perform a softmax operation on the aggregated output, reducing the impact of scale of the noise. Then, the obtained output is corrected by taking an average weighted by a parameter $\gamma$ and according to equation $\Phi_i^{corrected} = \gamma \bar{\Phi}(AD) + (1- \gamma)\Phi_i(AD)$ (equation 3 of paper). We observe that by appropriately choosing $\gamma$, the $\Phi_i^{corrected}$ and the noise in it serves as a regularizer for the local neural network helping it generalize better specially in small data cases when only 50 data instances per class are present on each client. This is also seen in the experimental results in the paper where the private version of our method performs better than other baselines in small data settings, as also mentioned by the reviewer. Since, we restrict the overall privacy budget to be low (less than 10), we have to add rather higher amount of noise (standard deviation $\approx 100$), and to also achieve lower privacy loss per round of global communication (less than 0.1), for DP based experiments we keep communication rounds $E = 100$ and $K =800$. We also observe that the best value of $\gamma$ for the private version of the algorithm turns out be lower ($\approx 0.4$) than the best gamma in non-private versions ($\approx 0.7$) adjusting for the increased noise in the aggregated output.
>
> -- *"Minor: presentation of the hierarchy in Algorithm 1 can be improved."*
>
> We thank the reviewer for suggesting this. We have modified this in the revised version of the paper.
>
> [3] Bassily et al., Private Query Release Assisted by Public Data, ICML 2020
>
> [4] Bassily et al., Learning from Mixtures of Private and Public Populations, Neurips 2020

---

> > ### Author Response · Authors · 2023-11-22
> > **Regarding our response**
> >
> > Dear reviewer aEw6,
> >
> > We appreciate your feedback and have carefully addressed the concerns in our response, and made modifications to the revised paper. If you have any further questions or comments, kindly let us know, we would be happy to address them as well. Thank you!

---

### Official Review · Reviewer_HKw3 · 2023-10-31

**Soundness:** 2 fair
**Presentation:** 2 fair
**Contribution:** 1 poor
**Rating:** 5
**Confidence:** 4

**Summary:**

The authors propose a personalized FL approach based on aggregating scores on publicly available dataset instead of aggregating model parameters. They use a Bayesian framework to model the heterogeneity, consequently the optimization problem. Score aggregation, which is motivated by the Bayesian framework, enables collaboration of clients with different architectures. They also utilize DP to prevent information leakage due to sharing scores on public dataset. They provide experiments on various dataset and provide utility-privacy tradeoff of the algorithm.

**Strengths:**

- The method is architecture agnostic, enabling the collaboration of resource heterogeneous clients.
- Use of Bayesian modelling in formulating the problem is clearly explained.
- The authors use DP to reduce information leakage due to sharing scores on a public dataset.
- Presented experiments show significant performance increase compared to competing methods.

**Weaknesses:**

- The major weakness is that many of the ideas presented in the paper was already explored in the literature. And these papers are not discussed in the related works. Let me elaborate,

The idea of aggregating scores on a public dataset instead of model weights was presented in [1] in 2019 although for training a global model. Using a Bayesian view for modelling the heterogeneity problem in FL through learning a prior was studied in [2,3,4]. Especially, [2,4] should be compared to in details in terms of modelling, note [4] also uses DP for enhancing privacy. The idea of collaborating resource and data heterogenous clients is presented in the paper as a novel contribution, however, [1,5] already did it (again there is no comparison to those methods).

Overall, it is not clear what is the contribution of the paper over those papers since they are not mentioned in the related works.

- Public unlabeled dataset is seen as a mild assumption. But, imagine planting this additional dataset to millions of devices, this can introduce inefficiencies in the system and waste client's precious resources.

- Generating $\Phi^{corrected}$ requires tuning an additional hyper-parameter $\gamma$ which further introduces inefficiency, since each client might need to do validation runs to tune it.

- There are several unclear points to me in the experiments. For instance what is the overall privacy budget for your DP method and DP-FedAvg, it says epsilon per round is smaller than 0.1 but what is the exact budgets?

- The majority of the experiments in the main text is done using 20 clients. It is not clear to me if the performance difference will persist in higher client regime, which is more realistic for FL.

- There is no comparison to local only training, this is a critical point.

- The supplementary material does not include code.

[1] Li et al., FedMD: Heterogenous Federated Learning via Model Distillation, https://arxiv.org/pdf/1910.03581.pdf

[2] Kotelevskii et al., FedPop: A Bayesian Approach for Personalised Federated Learning,
 https://proceedings.neurips.cc/paper_files/paper/2022/hash/395409679270591fd2a70abc694cf5a1-Abstract-Conference.html

[3] Chen et al., Self-Aware Personalized Federated Learning
, https://openreview.net/forum?id=EqJ5_hZSqgy

[4] Ozkara et al., A Statistical Framework for Personalized Federated Learning and Estimation: Theory, Algorithms, and Privacy, https://openreview.net/forum?id=FUiDMCr_W4o

[5] Ozkara et al., QuPeD: Quantized Personalization via Distillation with Applications to Federated Learning,  https://openreview.net/forum?id=Yowoe1scJOD

**Questions:**

Please address the points above.

---

> ### Author Response · Authors · 2023-11-17
> **Response to Reviewer HKw3 (1/2)**
>
> We thank the reviewer for the comments. We are glad that the reviewer finds that for our approach "Use of Bayesian modelling in formulating the problem is clearly explained" and that "Presented experiments show significant performance increase compared to competing methods."
>
> We address each of the reviewer's comments below.
>
> -- *"The major weakness is that many of the ideas presented in the paper was already explored in the literature. And these papers are not discussed in the related works."*
>
> The reviewer's main concern is that many of the ideas have been explored in the literature but have not been compared with in the paper, and provides us with 5 prior works that explored similar ideas. In the following, we discuss how our work is different from the previous works.
>
> First, our problem is different. We consider a specific problem of enabling efficient federated learning of Bayesian neural networks under client heterogeneous and data heterogeneous settings. This setting has not been explored in any of the prior works [1-5].
>
> Though the methods in [1] and [5] do propose knowledge distillation for collaboration amongst clients in FL settings to be able to train heterogeneous models, their algorithms are designed to work in the settings when we have point estimates of the parameters and not for the Bayesian models. Bayesian learning based parameter estimation is important in settings when we have very limited data or when we are trying to obtain better uncertainty estimates, it is however non-trivial to extend the knowledge distillation framework for BNNs. While some past works have explored knowledge distillation for BNNs, they either match posterior distributions [6] which requires identical parameters across BNNs or assume a simplistic prior [7] on the weights. In this work we propose a better and efficient way of knowledge distillation in heterogeneous BNNs (of different architectures) by way of assigning priors via the output space.
>
> Similarly, the methods in [2,4] assume a hierarchical statistical model for personalised FL across clients. In both those papers, the hierarchical model constitutes of an unknown population parameter also referred to as prior (denoted by $\beta$ in [2] and $\Gamma$ in [4]) from which the individual local parameters for each client are sampled (denoted by $\{ \theta_1, \theta_2, .. \}$ in [4] and $\{ z_1, z_2, .. \}$ in [2]) where the variance of the population parameter determines the heterogeneity of the client specific parameters. *Since the client's local parameters are sampled from the same distribution (also called prior), they have identical shape and form and therefore these methods work only in settings when the clients architectures are homogeneous*. This is also evident in the aggregation mechanisms used in the algorithms of [2,4] where they perform element wise aggregation over the parameters.
>
> While [3] also works in the homogeneous architecture settings, the method described in [3] does not use Bayesian learning based neural networks. [3] uses a Bayesian model to generate relevant insights about the variance and uncertainty in the local and global models and then uses those insights to determine local model initializations, computing the weighted aggregates of the local models, the local learning rate, number of local epochs for each client etc.
>
> In summary, our work is different from the suggested prior works and we believe that we provide a novel solution to a useful problem. We have also added the suggested papers as references and included them in related work.
>
> [1] Li et al., FedMD: Heterogenous Federated Learning via Model Distillation
>
> [2] Kotelevskii et al., FedPop: A Bayesian Approach for Personalised Federated Learning
>
> [3] Chen et al., Self-Aware Personalized Federated Learning
>
> [4] Ozkara et al., A Statistical Framework for Personalized Federated Learning and Estimation: Theory, Algorithms, and Privacy
>
> [5] Ozkara et al., QuPeD: Quantized Personalization via Distillation with Applications to Federated Learning
>
> [6] Shen et al., Variational Learning of Bayesian Neural Networks via Bayesian Dark Knowledge
>
> [7] Vadera et al., Generalized Bayesian Posterior Expectation Distillation for Deep Neural Networks

---

> ### Author Response · Authors · 2023-11-17
> **Response to Reviewer HKw3 (2/2)**
>
> We address the other comments from the reviewer as follows:
>
> -- *"Public unlabeled dataset is seen as a mild assumption. But, imagine planting this additional dataset to millions of devices, this can introduce inefficiencies in the system and waste client's precious resources."*
>
> Having the public unlabeled dataset could increase the memory footprint on the device, we agree, but as shown in Figure 3 in Appendix the number of data points required in the alignment dataset is not a lot, i.e., only about 2000 data points give a desired performance at the much reduced communication cost as compared to the conventional methods sharing millions of model parameters. Moreover, in cross-silo FL settings like hospitals, organizations etc. this additional memory may be negligible. We also note that several other methods have proposed enhancements in FL and privacy by using an auxiliary dataset, example 1,8,9,10 etc..
>
> [8] Sattler et al., FedAUX: Leveraging Unlabeled Auxiliary Data in Federated Learning
>
> [9] Goodfellow et al., Semi-supervised Knowledge Transfer for Deep Learning from Private Training Data
>
> [10] Chen et al., FEDBE: MAKING BAYESIAN MODEL ENSEMBLE APPLICABLE TO FEDERATED LEARNING
>
> -- *"Generating $\phi^{corrected}$ requires tuning an additional hyper-parameter $\gamma$ which further introduces inefficiency, since each client might need to do validation runs to tune it."*
>
> Tuning $\gamma$ is done at the beginning of the learning process for our method and does not require to be updated in each global round. Yet tuning it along with other hyper-parameters like learning rate, optimizer etc. might not be an additional problem for some clients, especially so in cross-silo settings, but even for smaller capacity clients we could think of empirical ways to obtain $\gamma$ as a function of the distance between the local and global representations. Also, in very low compute devices the server can also help the clients in tuning the gamma since server has access to both local representations and global representation.
>
> -- *"There are several unclear points to me in the experiments. For instance what is the overall privacy budget for your DP method and DP-FedAvg, it says epsilon per round is smaller than 0.1 but what is the exact budgets?"*
>
> We did mention it in the end of the privacy analysis section that our $\delta = 10^{-4}$ and $\epsilon$ is single digit but we make modifications to paper and include the value of $\epsilon \approx 9.9$.
>
> -- *"The majority of the experiments in the main text is done using 20 clients. It is not clear to me if the performance difference will persist in higher client regime, which is more realistic for FL."*
>
> The choice of the datasets and experiment settings is primarily motivated by the experiment settings in the baseline method papers and to show the general efficacy of the proposed method. We also have additional experiments in the Appendix in Table 4 with much more, 500 number of clients, and we see increased performance there as well. The cross-silo FL setting also typically involves less than 100 clients.
>
> -- *"There is no comparison to local only training, this is a critical point."*
>
> While we included this in Table 2 of the Appendix for the comparison in specific settings, we thank the reviewer for suggesting this. We have now added it in the experiments in Table 1 of the main paper.
>
> -- *"The supplementary material does not include code."*
>
> We ensure the reviewer to release the code upon acceptance of the paper.
>
> We have also revised the paper to include reviewer's suggestions and all the updates are in the blue color for clarity.

---

> ### Author Response · Authors · 2023-11-21
> **Regarding our response**
>
> Dear Reviewer HKw3,
>
> We appreciate your feedback and have carefully addressed the concerns in our response, and made modifications to the revised paper. If you have any further questions or comments, kindly let us know, we would be happy to address them as well.
>
> Thank you!

---

> > ### Comment · Reviewer_HKw3 · 2023-11-21
> > **Thanks for the rebuttal**
> >
> > Dear Authors,
> >
> > Thanks for the detailed rebuttal. You have addressed many of my concerns. However, my two main concerns did not completely vanish. I understand that your main contribution is based on introducing Bayesian NN/inference in the data and resource heterogeneous setting, but I do not think this is enough contribution to surpass the bar for acceptance (since this setting was studied before). Secondly, as you indicated in your rebuttal, most of the experiments are for cross-silo setting with the exception of Table 4. I believe this weakens the message conveyed by the experiments; and I think experiments with higher number of clients should be added for all the datasets.
> >
> > Overall, I increase my score to borderline reject; as I do not see enough evidence for the acceptance of the paper.
> >
> > Thanks.

---

> ### Author Response · Authors · 2023-11-21
> **Clarifications to reviewer's post-rebuttal response**
>
> We thank the reviewer for responding to the rebuttal.
>
> However, we would like to clarify again that the Bayesian inference in resource and data heterogeneous FL settings **has not** been studied before, as also explained in the rebuttal above and the paper. In fact, one of the primary contribution of this work is identification of need for such a setting and the bulk of our efforts are towards developing a method that indeed works in such a setting.
>
> We demonstrate that our method significantly outperforms baselines in various types of data, compute and statistical heterogeneous settings via extensive experiments, as also noted by other reviewers. And as mentioned in the previous comments, the choice of the datasets and experiment settings is primarily motivated by the experiment settings in the baseline method papers and to show the general efficacy of the proposed method.
>
> Both the set of experiments, including extensive set of experiments in cross-silo settings and additional experiments in cross-device setting, show an increased performance over the baseline methods. Therefore, the message from both sets of experiments is similar. We further assure the reviewer that given our existing setup for the cross-device experiments, with more time (since the runs for cross-device settings take much longer) we will be able to add those experiments for all datasets.

---

### Author Response · Authors · 2023-11-20
**Summary of Updates**

We thank all the reviewers for their valuable comments and suggestions which helped us revise the paper. We revised the paper based on reviewers' suggestions and highlight the modified parts in the blue color. We also provide clarifications to specific reviewer questions and concerns below as official comments along with experimental evidence when required.

The key updates in the paper are as follows :

1. Adding a new section discussing Bayesian Learning in Appendix A.

2. Adding results for Local Training as baseline in the main paper under all the settings.

3. Adding additional references and comparing to suggested works in the related work section.

4. Adding a non-Bayesian baseline based on knowledge distillation based collaboration for personalized FL (that represents a non-Bayesian version of the proposed algorithm), as suggested by reviewers.

5. Modifying the algorithm to improve readability.

---

### Meta-Review · Area_Chair_HaVn · 2023-12-14

**Metareview:**

This work proposes a federated learning based on Bayesian modeling and the distillation based approach from [1]. In this approach it is assumed that there is a public unlableled dataset and clients and server communicate labels for this dataset instead of model parameters in the more standard setup. This approach is significantly more restrictive as it requires a dataset from the same distribution. At the same time it is somewhat easier to implement with privacy and is also particularly relevant in a cross-silo FL where clients might use different architectures. In addition to describing this new combination of existing techniques this work gives an empirical evaluation that compares with multiple baselines in a variety of settings. While the experimental part of the work suggests that this combination of techniques is promising, most of the comparison is with techniques that do not have access to a public dataset. The most relevant comparison with [1] and other distillation based techniques was missing in the initial version. Comparison with [1] was added in the review process but even then with minimal level of detail and ablations (which are necessary given a large number of synthetic experiment design choices in the whole setup). Without a detailed apples-to-apples comparison I cannot recommend acceptance.

[1] Li et al., FedMD: Heterogenous Federated Learning via Model Distillation, https://arxiv.org/pdf/1910.03581.pdf

**Justification For Why Not Higher Score:**

see above

**Justification For Why Not Lower Score:**

n/a

---

### Decision · Program_Chairs · 2024-01-16

Reject